# MBNL1 regulates essential alternative RNA splicing patterns in MLL-rearranged leukemia

Svetlana S. Itskovich[1,9], Arun Gurunathan [2,9], Jason Clark [1], Matthew Burwinkel[1], Mark Wunderlich[3], Mikaela R. Berger[4], Aishwarya Kulkarni[5,6], Kashish Chetal[6], Meenakshi Venkatasubramanian[5,6], Nathan Salomonis [6,7], Ashish R. Kumar [1,7] & Lynn H. Lee [7,8✉]

Despite growing awareness of the biologic features underlying MLL-rearranged leukemia, targeted therapies for this leukemia have remained elusive and clinical outcomes remain dismal. MBNL1, a protein involved in alternative splicing, is consistently overexpressed in MLL-rearranged leukemias. We found that MBNL1 loss significantly impairs propagation of murine and human MLL-rearranged leukemia in vitro and in vivo. Through transcriptomic profiling of our experimental systems, we show that in leukemic cells, MBNL1 regulates alternative splicing (predominantly intron exclusion) of several genes including those essential for MLL-rearranged leukemogenesis, such as *DOT1L* and *SETD1A*. We finally show that selective leukemic cell death is achievable with a small molecule inhibitor of MBNL1. These findings provide the basis for a new therapeutic target in MLL-rearranged leukemia and act as further validation of a burgeoning paradigm in targeted therapy, namely the disruption of cancer-specific splicing programs through the targeting of selectively essential RNA binding proteins.

[1] Division of Bone Marrow Transplantation and Immune Deficiency, Cincinnati Children's Hospital Medical Center, Cincinnati, OH 45229, USA. [2] Cancer and Blood Diseases Institute, Cincinnati Children's Hospital Medical Center, Cincinnati, OH 45229, USA. [3] Division of Experimental Hematology and Cancer Biology, Cincinnati Children's Hospital Medical Center, Cincinnati, OH 45229, USA. [4] College of Medicine, University of Cincinnati School of Medicine, Cincinnati, OH 45267, USA. [5] Department of Electrical Engineering and Computer Science, University of Cincinnati, Cincinnati, OH 45221, USA. [6] Division of Biomedical Informatics, Cincinnati Children's Hospital Medical Center, Cincinnati, OH 45229, USA. [7] Department of Pediatrics, University of Cincinnati School of Medicine, Cincinnati, OH 45229, USA. [8] Division of Oncology, Cincinnati Children's Hospital Medical Center, Cincinnati, OH 45229, USA. [9] These authors contributed equally: Svetlana S. Itskovich, Arun Gurunathan. ✉email: lynn.lee@cchmc.org

Reciprocal chromosomal translocations of the mixed lineage leukemia (MLL) gene with one of over 70 fusion partners are commonly found in a subset of aggressive leukemias including infant leukemia, and are associated with poor outcome[1–3]. Regardless of the fusion partner and the type of leukemia (acute myeloid leukemia (AML) or acute lymphoblastic leukemia (ALL)) all MLL-rearranged leukemias share a common gene expression profile that is distinct from that of non−MLL-rearranged leukemias[4,5]. Analysis of MLL-rearranged signatures reveals that muscleblind-like 1 (MBNL1) is one of the most consistently overexpressed genes in MLL-rearranged leukemia compared to other leukemias[3–6].

Muscleblind-like 1 (MBNL1) is an RNA-binding protein that has been shown to regulate RNA alternative splicing, localization, and integrity[7–11]. Alternative splicing of pre-mRNA is a key mechanism regulating eukaryotic gene expression by expanding genome coding diversity. MBNL1 has previously been shown to regulate alternative splicing in the fetal-to-adult transition in heart, muscle, and brain[9]. Sequestration of MBNL1 due to binding to expanded poly-CUG repeats in the dystrophia myotonica protein kinase (DMPK) mRNA is a main contributor to the pathogenesis of myotonic dystrophy type 1 (DM1), a syndrome characterized by muscle weakness, cardiac abnormalities, intellectual disability, and cataracts. The sequestration of MBNL1 results in missplicing of target mRNAs, leading to the clinical features[12]. Moreover, MBNL1 and MBNL2 are known to play a central role in regulating the pattern of alternative splicing events that control embryonic cell pluripotency[13]. Knockdown of Mbnl1 in murine fetal liver cells leads to blockade of terminal erythropoiesis through deregulation of alternative splicing of Ndel1 mRNA[14].

Abnormalities in RNA processing, particularly splicing, have been associated with cancer, including AML and ALL)independent of spliceosome mutations[15–19]. The increased expression of MBNL1 in MLL-rearranged leukemia described above, as well as evidence that the MLL-fusion complex directly binds the MBNL1 promoter[20], suggest that MBNL1-mediated RNA splicing may be important to the pathogenesis of MLL-rearranged leukemias. The mechanism of action and extent of essentiality of MBNL1 in MLL-rearranged leukemogenesis, however, remains unknown.

In this report, through a combination of functional genomic studies, pharmacologic inhibition, and comprehensive analysis of alternative splicing we demonstrate that MBNL1 is required in MLL-rearranged leukemia.

## Results

**MBNL1 is required for the propagation of human MLL-rearranged leukemia in vitro and in vivo.** To confirm the relevance of MBNL1 expression in MLL-rearranged leukemia, we first compared multiple gene expression studies which identified differentially expressed genes between MLL-rearranged and MLL-wildtype leukemias[4,6,21,22]. We began by examining the intersection of these gene expression signatures, and found that MBNL1 expression was a common feature of these signatures across both acute myeloid and lymphoblastic leukemias (Fig. 1a). We further examined MBNL1 expression across two major primary patient datasets evaluating both AML and ALL, and consistently found high MBNL1 expression in MLL-rearranged patient samples[23–28] (Supplementary Figs. 1A–C). We subsequently analyzed expression levels of MBNL1 by quantitative RT-PCR (qRT-PCR) in human leukemia cell lines. We found MBNL1 expression in all leukemic cell lines tested, with the highest MBNL1 expression in MLL-rearranged cell lines (Fig. 1b). Additionally, human CD34+ cord blood transformed with the MLL-AF9 (MA9) oncogene demonstrated higher levels of

MBNL1 expression compared to normal CD34+ cord blood cells (Supplementary Fig. 1D). A similar phenomenon was observed in Lin- mouse bone marrow cells transduced with the MA9 retrovirus (Supplementary Fig. 1E) MBNL1 was recently demonstrated to be a direct target of the MLL-AF4 fusion protein in patient-derived ALL cell lines as well as in an experimental retroviral model[20,29]. To determine whether this observation applied to leukemia cells bearing different MLL-fusion partners, we analyzed MLL-fusion protein (MLL-N and fusion partner C-terminus if applicable) and H3K79me2 chromatin immunoprecipitation followed by deep sequencing (ChIP-seq) datasets from THP-1[30] and ML-2[31] cell lines (with MLL-AF9 and MLL-AF6 respectively), as well as from the MV4;11[30], RS4;11[32], and SEM[33] cell lines which bear an MLL-AF4 fusion. We found evidence of MLL-N/fusion-C binding to the MBNL1 promoter and gene body, along with H3K79me2 enrichment across all cell lines studied (Fig. 1c). To experimentally verify the observed interactions between MLL-fusion proteins and MBNL1, we used transformed human CD34+ cells bearing a repressible MA9 oncogene where doxycycline treatment represses MA9 expression[34,35]. Nuclear extracts isolated at different intervals demonstrated a decrease in MBNL1 expression directly correlating with MA9 downregulation (Fig. 1d).

We then performed short hairpin RNA (shRNA) knockdown studies of MBNL1 in human leukemia cell lines to test its requirement for leukemia cell growth. We screened commercially available lentiviral shRNAs and chose two that showed the most consistent and efficient knockdown of MBNL1 mRNA (Supplementary Fig. 2A) and protein (Fig. 2a), hereafter referred to as shMBNL1-64 and -65. There was substantial growth inhibition of MLL-rearranged cell lines following MBNL1 knockdown both in liquid culture (Fig. 2b–d) and in colony forming unit (CFU) assays (Supplementary Figs. 2B–C). However, MBNL1 knockdown did not significantly affect growth of non-MLL-rearranged leukemic cells (Fig. 2e–g) bearing a variety of different oncofusions (BCR-ABL in K562, RUNX1-RUNXT1 in Kasumi-1) despite efficient knockdown (Supplementary Fig. 2D–H).

To determine the requirement for MBNL1 in leukemia in vivo, we transduced two primary patient AML samples bearing MLL fusions (one MLL-AF9 and one MLL-AF10) with shMBNL1-64 or a non-targeting (NT) control. Transduction efficiency ranged from 67 to 85% (Venus+). Bulk cells were transplanted into immune-deficient NSGS mice. At the time of visible illness, all mice showed robust human cell engraftment (ranging from 92.4 to 99.4% in the MLL-AF9 group and 7.6–40.2% in the MLL-AF10 group) in the bone marrow. Bone marrow from mice transplanted with shNT-transduced cells contained Venus+ cells (range 6.9–78% of all human cells), whereas the majority of mice transplanted with shMBNL1-64 had virtually no Venus+ fraction (Fig. 2h). One mouse transplanted with shMBNL1-64-transduced MLL-AF9 patient cells had 62.4% Venus+; this mouse was removed from analysis after qRT-PCR analysis revealed lack of MBNL1 knockdown. To show broad applicability of knockdown, we repeated this experiment using an MLL-AF9/NRAS^G12D cell line, with similar results (Supplementary Fig. 3A–C)[36]. These data suggest that MBNL1 is essential for leukemia propagation in vivo.

As an alternative approach, we tested the effects of MBNL1 inhibition achieved by biochemical means. Recently, an MBNL1-specific inhibitor that prevents MBNL1 binding to its targets was identified in a high-throughput screen for compounds with activity in myotonic dystrophy[37]. We tested the effect of this compound on MLL-rearranged (MV4;11, MOLM13) and non-MLL-rearranged leukemia cell lines (HL-60, K562, Kasumi-1), as well as on normal CD34+ cord blood cells. Cells were incubated with the inhibitor at various concentrations (based on previous experiments). Leukemia cells were sensitive to the inhibitor

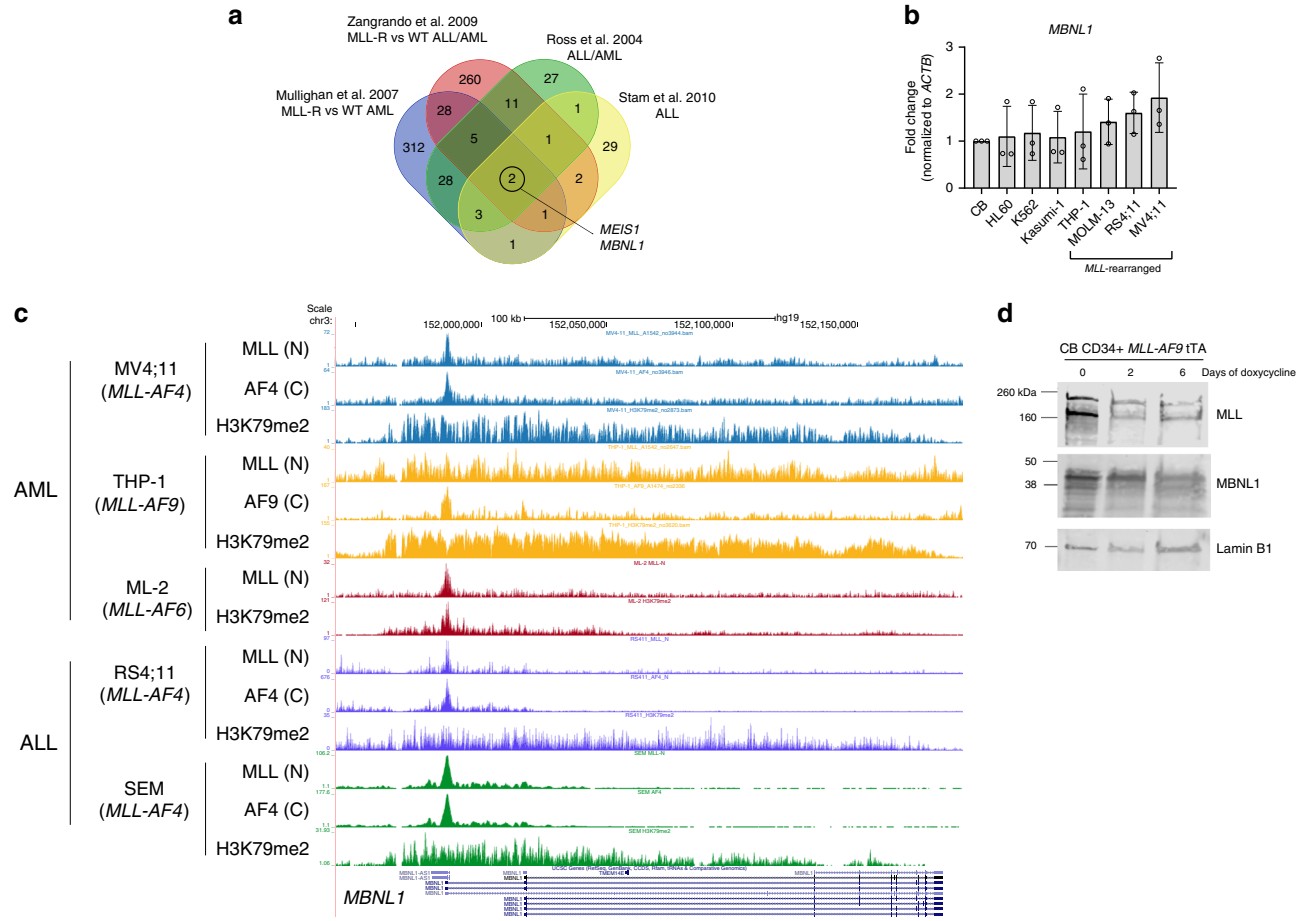

**Fig. 1 MBNL1 is overexpressed in MLL-rearranged leukemias and MLL-fusion proteins interact with MBNL1. a** Intersection of published gene expression signatures composed of genes overexpressed in MLL-rearranged AML and ALL when compared to other MLL-wildtype leukemias. **b** Relative expression of *MBNL1* in non-MLL-rearranged (Kasumi-1, HL60, and K562) cell lines and MLL-rearranged (THP-1, RS4;11, MOLM13 and MV4;11), normalized to CD34+ cord blood expression. Data is from three biological replicates. Bars show mean ± SD. **c** ChIP-seq tracks of human AML and ALL cell lines expressing different MLL-fusion proteins. ChIP-seq data were obtained from GSE95511 for ML-2, GSE79899 for MV4;11 and THP-1, GSE38403 for RS4;11, and GSE38338 for SEM. **d** Western blot analysis of MBNL1 levels in *MLL-AF9* Tet-off human CD34+ cells. Nuclear protein levels were analyzed on day 0, 2, and 6 after doxycycline treatment. Lamin B1 was used as a loading control. Representative western blot shown, two biological replicates performed.

treatment as evidenced by a decrease in cell viability within 18 h in a variety of leukemia cell lines. Despite the high concentrations necessary to achieve the observed effects, MLL-rearranged leukemia cells demonstrated comparatively higher sensitivity to the compound (Fig. 2i, j).

**MLL-AF9 leukemia is delayed in *Mbnl1* knockout (KO) mice.** As a complementary means of characterizing the essentiality of *MBNL1* in MLL-rearranged leukemia, we used a mouse model of *Mbnl1* genetic ablation. For these experiments we used $Mbnl1^{\Delta E3/\Delta E3}$ ($Mbnl1^{-/-}$) KO mice lacking *Mbnl1* exon 3 (kindly provided by Dr. Maurice Swanson)[38]. We transduced $Mbnl1^{-/-}$ and littermate control ($Mbnl1^{+/+}$) lineage-negative bone marrow cells with an MLL-AF9 GFP-bearing retrovirus[34]. Four days after transduction, $Mbnl1^{+/+}$ and $Mbnl1^{-/-}$ cells were plated in semi-solid culture for at least two weekly serial rounds to select transformed cells, which were subsequently transplanted into syngeneic irradiated mice (Fig. 3a). Moribund animals were classified as having developed leukemia at necropsy based on the presence of splenomegaly, elevated white blood cell count in the peripheral blood, and leukemic cell infiltration in the bone marrow and spleen. In these experimental settings, mice transplanted with transformed $Mbnl1^{+/+}$ cells developed leukemia with a median time of 84 days. However, mice transplanted with

transformed $Mbnl1^{-/-}$ cells demonstrated a significantly prolonged latency and leukemia-free survival of 123 days after transplantation (Fig. 3b). Both groups showed comparable levels of splenomegaly, high white blood cell counts, anemia, and thrombocytopenia. To characterize the influence of *Mbnl1* loss on leukemia-initiating cells in vivo we performed secondary transplantation into irradiated hosts using splenocytes from sick primary mice. Similar to our primary transplant experiments, secondary recipients of $Mbnl1^{-/-}$ leukemia cells demonstrated longer leukemia-free survival (65 days) compared to mice secondarily transplanted with $Mbnl1^{+/+}$ leukemia cells (49 days) (Fig. 3c). In vitro studies of the CFU activity of $Mbnl1^{-/-}$ MLL-AF9-transformed cells from bone marrow or spleen of primary or secondary recipients in serial replating assays showed no significant difference in colony numbers compared to $Mbnl1^{+/+}$ MLL-AF9-transformed cells. These results indicate that while the retrovirally expressed MLL-AF9 fusion gene can initiate leukemia in the absence of *Mbnl1*, disease development is significantly delayed.

**Loss of *Mbnl1* has a modest effect on normal murine hematopoiesis.** In order to determine the role of Mbnl1 in normal murine hematopoiesis, we studied $Mbnl1^{-/-}$ mice. At steady state, $Mbnl1^{-/-}$ mice and $Mbnl1^{+/+}$ controls demonstrated

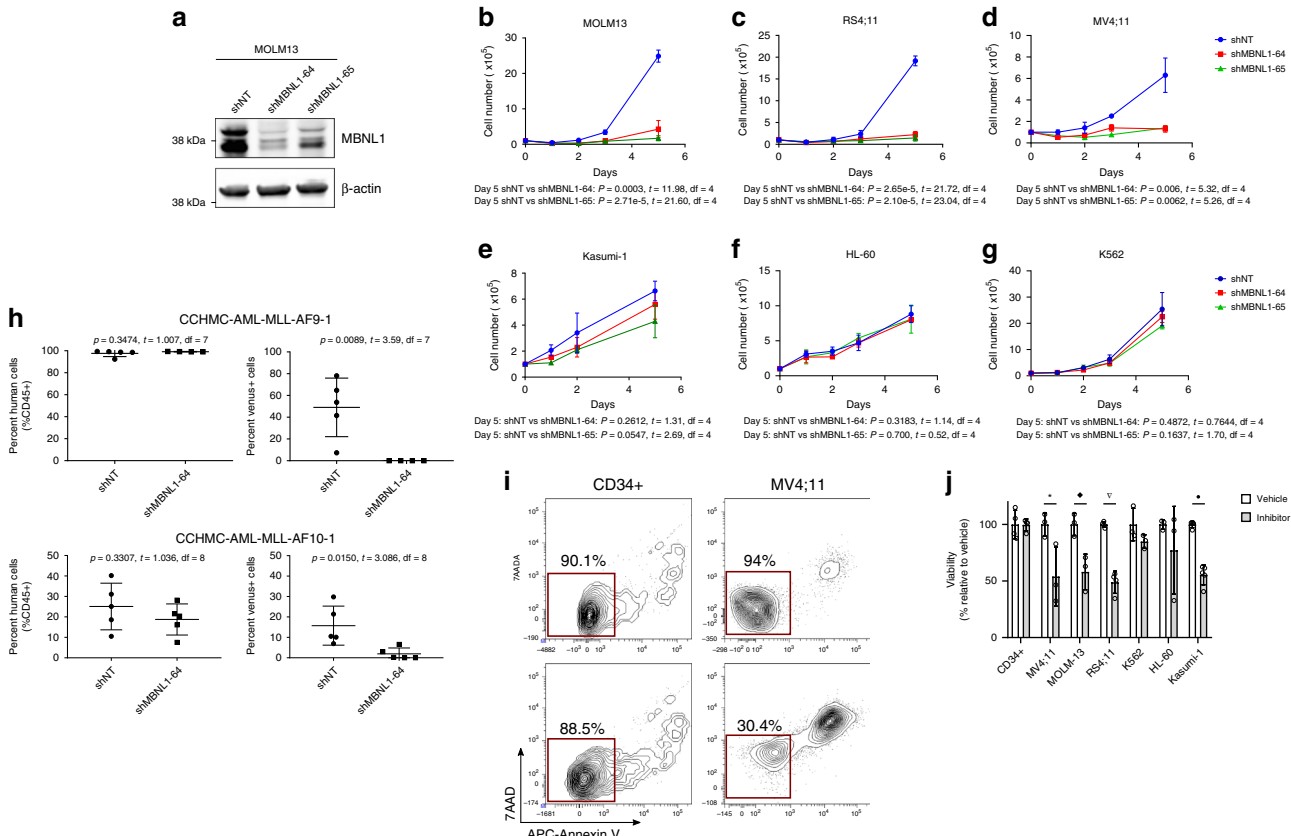

**Fig. 2 MBNL1 is required for the propagation of human MLL-rearranged leukemia in vitro and in vivo. a** Protein-level effect of *MBNL1* knockdown using two different shRNAs (shMBNL1-64, shMBNL1-65). Representative western blot shown, two biological replicates performed. **b–g** In vitro growth of leukemia cells upon shRNA knockdown of *MBNL1*. Data is from three technical replicates. Data shown for MOLM13 (**b**), RS4;11 (**c**), MV4;11 (**d**), Kasumi-1 (**e**), HL-60 (**f**), and K562 (**g**) cells. Day 0 refers to the day transduced cells were isolated by sorting. Cell growth was assessed by counting viable cells using Trypan Blue. Data represent mean ± SD. *MBNL1* knockdown was confirmed through qRT-PCR (Supplementary Fig. 2A, D–H). **h** Results (scatter plots with mean ± SD) of transplant of non-targeting (NT) and *MBNL1* knockdown (MBNL1-64) transduced MLL-AF9 and MLL-AF10 primary patient cells. Bone marrow cells were collected from mice (n = 4 mice in MLL-AF9 shMBNL1-64 group, n = 5 mice in all other groups) at time of leukemic illness. The plots on the left show percentage of human CD45$^+$ cells and the plots on the right show percentage of Venus-positive cells in CD45 + fraction. P values represent comparison between shNT and shMBNL1 conditions by unpaired two-tailed *t*-test. **i** CD34$^+$ cord blood cells and MV4;11 human leukemia cells were treated with vehicle or 500 μM small molecule inhibitor for 18 h and assessed for apoptosis. Representative density plots show flow cytometry analysis for Annexin V and 7-AAD staining gating on viable cells. **j** Graph shows relative cell viability by flow cytometry (Annexin V/7AAD) 18 h after inhibitor versus vehicle treatment in different cell lines. Viability normalized to 100% in vehicle-treated cells. Data represents 3–5 biological replicates. Error bars show mean ± SD. *p = 0.0465; ◆p = 0.0173; ∇p = 5.47e−5; ●p = 4.57e−6 by unpaired two-tailed *t*-test comparing vehicle vs. inhibitor for each cell line.

comparable peripheral blood counts, bone marrow cellularity, and progenitor and stem cell counts. To test the role of Mbnl1 in stress hematopoiesis we injected *Mbnl1*$^{−/−}$ KO and *Mbnl1*$^{+/+}$ wild type (WT) mice with one sublethal dose of 5-Fluorouracil (5-FU). No difference was observed regarding the time to recovery of RBCs, platelets, and WBCs from *Mbnl1*$^{−/−}$ mice compared to controls (Fig. 4a). Next we tested the ability of bone marrow cells to repopulate lethally irradiated recipients as a function of stem cell activity in serial (non-competitive) transplantations. Analysis of bone marrow and peripheral blood post-transplantation did not reveal any significant difference in donor chimerism between primary recipients of *Mbnl1*$^{+/+}$ or *Mbnl1*$^{−/−}$ bone marrow (Fig. 4b–d). Similarly, the distribution of cell lineages in the peripheral blood were identical between each group (Fig. 4e). Analysis of the bone marrow in secondary recipients also revealed no difference in donor chimerism levels between *Mbnl1*$^{+/+}$ and *Mbnl1*$^{−/−}$ mice (Fig. 4c). Overall, these data demonstrate that the murine hematopoietic system functions normally under steady state conditions in the absence of Mbnl1.

To more rigorously compare *Mbnl1* knock out and wild type hematopoietic function in vivo we performed competitive

transplantation experiments. Lethally irradiated BoyJ recipient mice were transplanted with *Mbnl1*$^{−/−}$ or *Mbnl1*$^{+/+}$ bone marrow cells (CD45.2) along with CD45.1 competitor bone marrow cells in a 1:1 ratio. Four weeks after transplantation the repopulation ability of transplanted cells was tested by peripheral blood flow cytometry for CD45.1 vs 45.2. Bone marrow analyses showed 73.8% and 30.3% CD45.2 donor chimerism in mice transplanted with *Mbnl1*$^{+/+}$ or with *Mbnl1*$^{−/−}$ bone marrow cells, respectively (Fig. 4f). Periodic analysis of peripheral blood chimerism every 4 weeks showed a slight increase in donor chimerism in mice transplanted with *Mbnl1*$^{+/+}$ bone marrow cells while donor chimerism in mice transplanted with *Mbnl1*$^{−/−}$ bone marrow cells remained stable (Fig. 4g). Analysis of peripheral blood lineages showed a delay in the ability of *Mbnl1*$^{−/−}$ marrow to generate T-lymphocytes in competitive transplants while Gr1$^+$ myeloid cells repopulated to levels comparable to *Mbnl1*$^{+/+}$ (Fig. 4h). These data suggest a slight disadvantage of *Mbnl1*$^{−/−}$ cells under stress conditions of competitive transplant, with a specific deficiency in the ability of *Mbnl1*$^{−/−}$ marrow to generate T-lymphocytes during competitive transplantation. To determine the etiology of this observation, we

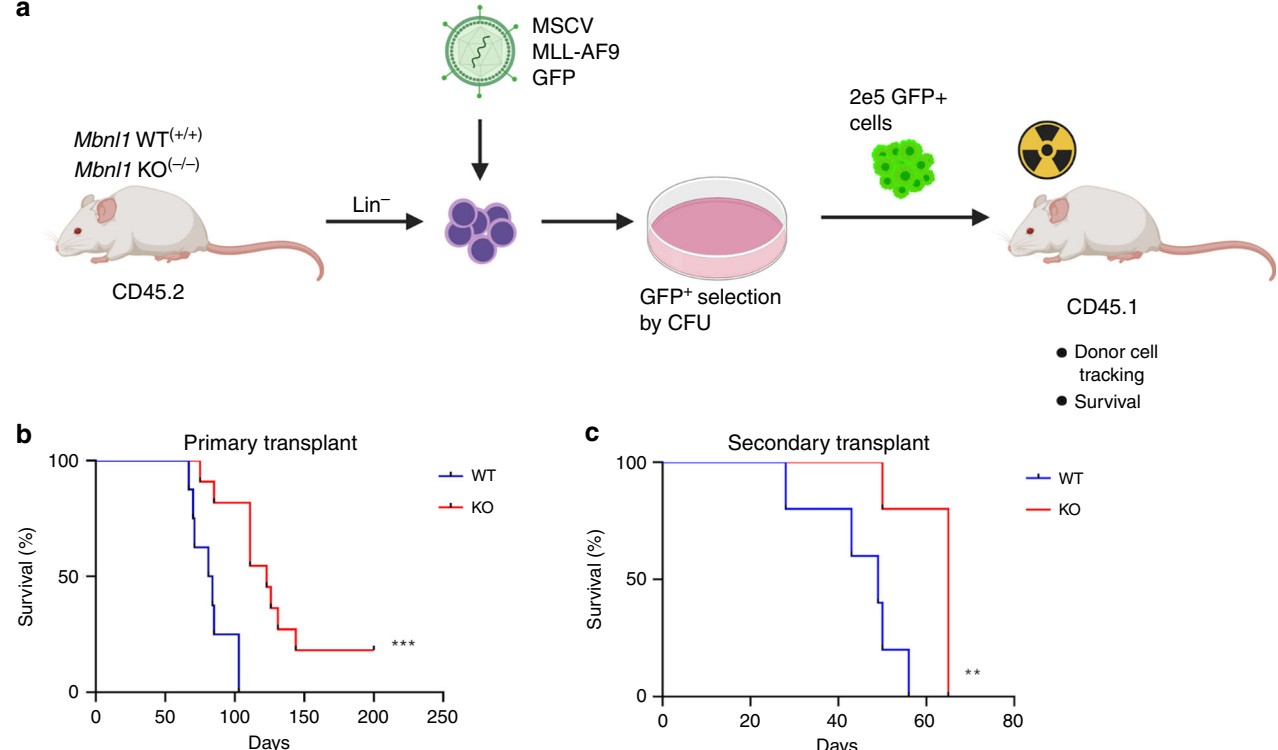

**Fig. 3 MLL-AF9 leukemia is delayed in Mbnl1 knockout mice. a** Scheme of mouse leukemia model. Wild type (WT) or knockout (KO) lineage negative bone marrow cells were transduced with an MLL-AF9 GFP-expressing retrovirus, followed by serial replating in semisolid medium to select for MLL-AF9 cells. Selected cells were then transplanted into irradiated primary recipient mice (n = 10 per group). **b** Graph represents percent of leukemia-free survival of primary recipient mice. ***p = 0.0003 by two-sided Log-rank (Mantel-Cox) test. **c** Distressed primary recipient mice 12 weeks after transplantation were sacrificed and splenocytes were harvested and plated for CFU assay. After 7 days in semi-solid culture spleen cells were injected into secondary irradiated recipients (n = 5 mice per group). Graph depicts percent of leukemia-free survival. **p = 0.0088 by two-sided Log-rank (Mantel-Cox) test.

characterized the HSC compartment of $Mbnl1^{-/-}$ mice and found that $Mbnl1^{-/-}$ mice had a decreased proportion of Lin⁻Sca1⁺Kit⁺-SLAM+(CD41-CD48-CD150+) HSCs compared to WT mice (Fig. 4i, j). Despite this, these mice at steady state exhibit no major hematopoietic deficits as evidenced by these experiments, suggesting that $Mbnl1$ loss is tolerated in normal hematopoiesis.

**MBNL1-regulated alternative splicing is active in leukemic cells**. One of the best known and explored functions of MBNL family proteins is alternative splicing of target mRNAs[7,13,39–41]. We sought to identify whether canonical MBNL1 targets exhibit differential patterns of isoform usage across leukemia cells with differing degrees of MBNL1 expression. We selected the genes $ATP2A1$ and $CD47$, which have been shown to be regulated by MBNL1 during pluripotent stem cell differentiation[41], as well as $INF2$, which is misspliced due to MBNL1 sequestration in the trinucleotide repeat disorder Fuchs endothelial corneal dystrophy[42]. In comparing the isoform expression profiles of these genes between MLL-wildtype and MLL-rearranged leukemias, we identified shifts in isoform utilization in MLL-rearranged leukemia cell lines which had high MBNL1 expression (Fig. 5a). Furthermore, $MBNL1$ knockdown in MLL-rearranged cell lines resulted in a shift in mRNA isoform expression resembling that of MBNL1-low cell lines (Fig. 5b)[41]. Pharmacologic inhibition of MBNL1 with Compound 1 as described above resulted in similar changes in alternative splicing of MBNL1 target genes (Fig. 5c). Intriguingly, these results suggest that at least some targets of MBNL1 are spliced in a tissue-agnostic manner, and that patterns of isoform expression associated with high $MBNL1$ expression are

reversed upon $MBNL1$ knockdown in leukemia cells. These data provide further evidence for on-target activity of our MBNL1 inhibitor.

As an additional measure of MBNL1 activity, we investigated differential exon utilization in $MBNL1$ itself. Specifically, MBNL1 promotes exclusion of exon 5 in its own mature transcripts[43]. We thus hypothesized that higher expression of $MBNL1$ will be associated with a greater proportion of $MBNL1$ transcripts with exon 5 excluded. Indeed, quantitative analysis revealed a higher ratio of exon 5 excluded transcripts in MLL-rearranged leukemias compared to non-MLL-rearranged leukemias (Supplementary Fig. 4A). Collectively, these data show that at least some putative targets of MBNL1 are consistent across different cell types, and that high $MBNL1$ expression correlates with increased alternative splicing of $MBNL1$ in MLL-rearranged leukemia.

**MBNL1 mediates intron retention of genes differentially spliced in MLL-rearranged leukemia**. To identify targets of MBNL1 specifically in MLL-rearranged leukemias, we performed RNA-Seq in two MLL-rearranged cell lines (MOLM13, MV4;11) with knockdown of $MBNL1$ and compared these results to those from the Leucegene consortium dataset of MLL-rearranged patient AML bone marrow samples[44]. For these analyses, we compared patient samples with MLL oncofusions to cytogenetically normal (CN) AML patients, and non-targeting shRNA controls to $MBNL1$ knockdown (NT). Similar to other AML datasets, we find that $MBNL1$ is most highly induced in MLL-rearranged AMLs in RNA-seq derived from the Leucegene cohort and comparable sorted healthy control hematopoietic progenitors (Fig. 6a).

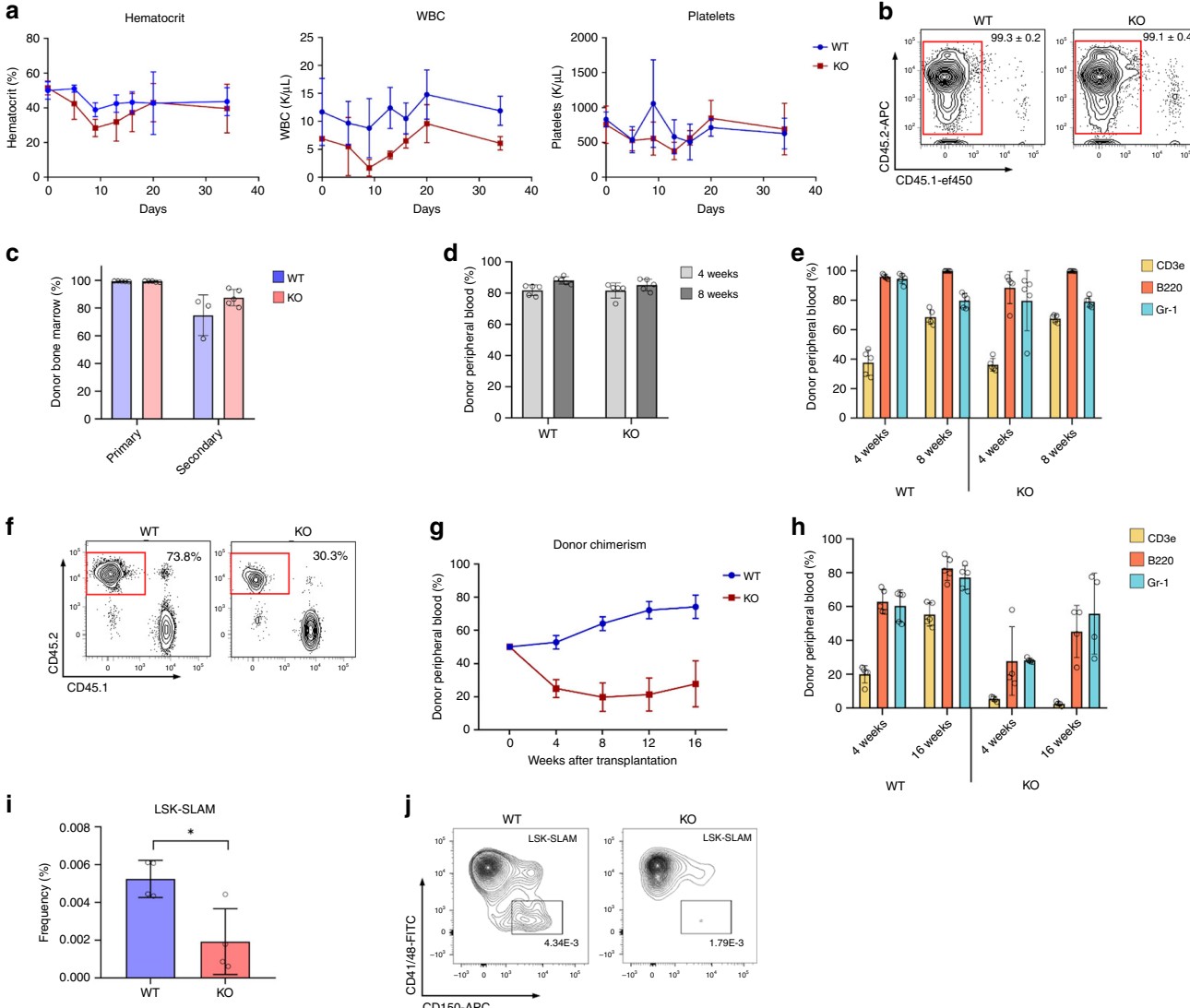

**Fig. 4 Loss of Mbnl1 has a modest effect on normal murine hematopoiesis. a** *Mbnl1* wild type (WT) and homozygous knockout (KO) mice (five mice per group) were treated with 5-Fluorouracil (150 mg/Kg), and assessed for the levels of hematocrit, WBC, and platelets every 4–5 days. Line graph represents mean ± SD. **b** Representative density plots show flow cytometry analysis of donor chimerism in bone marrow of primary recipients 12 weeks after non-competitive transplantation (gate overlaid on CD45.2-postive CD45.1-negative donor cells). **c** Donor chimerism in bone marrow of recipients of primary and secondary transplantation at 12 weeks post-transplant. Bars show mean ± SD ($n = 5$ mice per group). **d** Analysis of peripheral blood donor chimerism in primary recipients 12 weeks after non-competitive transplantation. Bars show mean ± SD ($n = 5$ mice per group). **e** Flow cytometry analysis of peripheral blood cells presentation in lymphoid (CD3e and B220) and myeloid (GR-1) lineages 4 or 8 weeks after non-competitive transplantation. Bars show mean ± SD ($n = 5$ mice per group). **f** Representative density plots show flow cytometry analysis of donor chimerism 12 weeks after competitive transplantation (gating on CD45.2-postive CD45.1-negative donor cells). **g** Graph represents peripheral blood donor chimerism after competitive transplantation over time (horizontal axis). Line graph represents mean ± SD ($n = 5$ mice per group). **h** Flow cytometry analysis of peripheral blood cells representation in lymphoid and myeloid lineages 4 or 16 weeks after competitive transplantation. Bars show mean ± SD ($n = 5$ mice per group). **i** LSK-SLAM + population in *Mbnl1* WT vs KO mice, represented as fraction of all analyzed cells. ($n = 4$ mice per group.) *$p = 0.0161$ by unpaired two-tailed *t*-test; error bars indicate mean ± SD. **j** Representative flow plots of LSK-SLAM + quantitation between experimental groups.

Notwithstanding the disparate experimental setups of primary patient samples versus knockdown in a cell line, we hypothesized that the major impact of MBNL1 might be on alternative splicing of transcripts, given its well defined role in splicing regulation. We applied a recently developed method for splicing event and intron-retention analysis called MultiPath-PSI (AltAnalyze) to the patient and *MBNL1* knockdown cell line data[45,46]. Breakdown of alternative splicing and promoter events in these datasets by event-type finds that MLL-rearranged leukemias predominantly exhibit exclusion of retained introns when compared to CN-AML (Fig. 6b, top). We observed a similar difference in MOLM13 cells

between MBNL1 knockdown and control, where loss of MBNL1 was predominantly associated with increased intron retention and a decrease in cassette exon inclusion (Fig. 6b, bottom). Relative to well-defined AML splicing-factor mutations and oncofusions in Leucegene, this intron-retention exclusion profile was most dominant in MLL-rearranged leukemias and to a lesser extent in core binding factor rearrangements (*RUNX1*, *CBFB-MYH11*) (Supplementary Fig. 5A) Visual inspection of AS events in *MBNL1* knockdown using IGV readily confirmed these bioinformatics predictions (Fig. 6c). Comparison of splicing changes occurring with MBNL1 knockdown to AS events

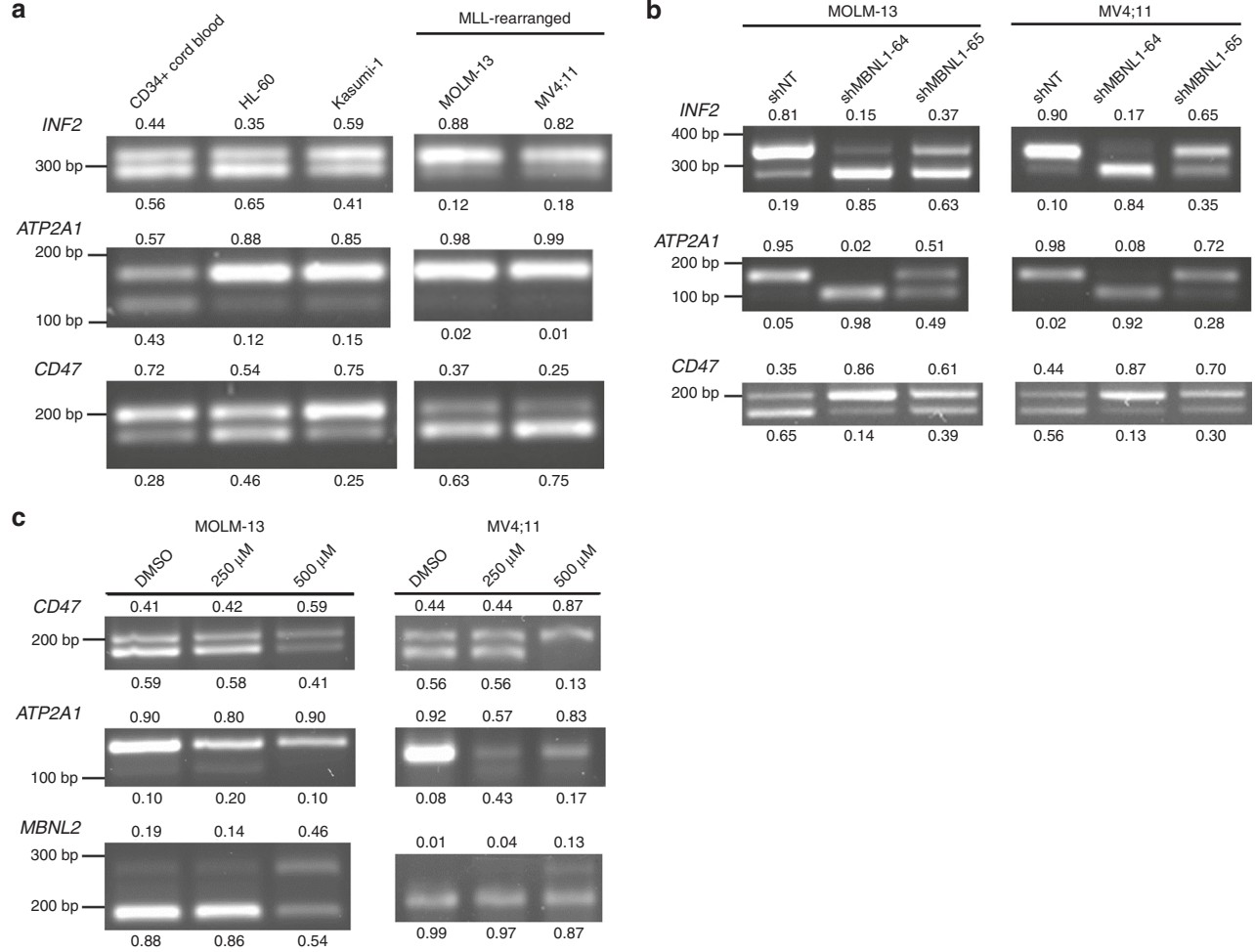

**Fig. 5 MBNL1-regulated alternative splicing is active in leukemic cells. a** RT-PCR analysis of regions of select MBNL1 regulated genes (*INF2, ATP2A1,* and *CD47*) across a panel of MLL-rearranged and wild-type leukemia cells. **b** RT-PCR analysis of *INF2, ATP2A1,* and *CD47* transcripts demonstrating change in splicing pattern in MOLM13 cells after *MBNL1* knockdown. Band percentages (relative to each individual lane) are noted. **c** Alternative splicing patterns of MBNL1 target genes are shown after 48 h of Compound 1 treatment at the indicated dosages. Relative proportion of each AS feature indicated in decimal adjacent to corresponding band. All gels are representative images of two biological replicates.

occurring in the MLL-rearranged patient cohort identifies a common core set of mutually regulated events. Intersection of these in vivo and in vitro comparisons finds that 89% of shared AS events are negatively concordant (e.g., oppositive pattern of exon/intron inclusion) between MLL-rearranged patient samples and in vitro *MBNL1* knockdown, namely that specific alternative features retained in MLL-rearranged patient data were excluded in the *MBNL1* knockdown condition, or vice versa. Multiple short hairpins against MBNL1 exhibited a high degree of concordance (>96%) in AS events across different cell lines (Fig. 6d, e; Supplementary Data 1). Manual inspection of these splicing events verified that MBNL1 knockdown rescues MLL fusion-associated splicing and intron retention events (Supplementary Fig. 6A, B). These data suggest that MBNL1 loss partially accounts for changes in specific AS events which occur in MLL-rearranged leukemias. Given that the number of in vivo and in vitro overlapping splicing events was relatively low, we more broadly examined concordance of MBNL1 knockdown AS events among major genetically/cytogenetically defined AML subtypes, including nearly 500 splicing factor knockdowns in K562 or HepG2 cells (ENCODE; Encyclopedia of DNA Elements) and normal hematopoetic cell types. Among these comparisons, MLL-rearranged AML proved to be the most highly anti-concordant to

*MBNL1* knockdown, with non-MLL rearranged *MBNL1* knockdown (in K562 cells) among the most concordant (Fig. 6f; Supplementary Data 2). While shRNAs for other RNA-binding proteins (RBPs) had concordant AS with MBNL1, suggesting they regulate common targets or are regulated downstream of MBNL1, binding sites for MBNL1 were the most enriched among these for *MBNL1* knockdown AS events, similar to that of MLL-rearranged splicing events (Fig. 6g, Supplementary Fig. 5B).

We subsequently examined the concordant AS decisions, and noted that they were highly enriched in regulators of small GTPase mediated signal transduction (*RHOC, RHPN1, DLG4, ARHGEF40, ARHGEF1, DENND4B, ARHGAP17, ARAP1, DNM2*), regulators of Notch signaling (*NCOR2, NUMB, AA2*) and genes mediating membrane trafficking and vesicle-mediated transport (*BICD2, REPS1, SCARB1, SEC16A, DENND4B, AAK1, AP1G2, EXOC1, EXOC7, GOLGA2, DNM2*) (Fig. 6h). Furthermore, we consistently identified missplicing (specifically intron retention) of key genes implicated in the pathogenesis of MLL-rearranged leukemia, namely *SETD1A* and *DOT1L* (Fig. 6h). Both DOT1L and SETD1A are histone methyltransferases known to perform essential functions in the MLL leukemogenic program[47–49]. We hypothesized that of the alternatively spliced targets we identified, disruptions of genes involved in the

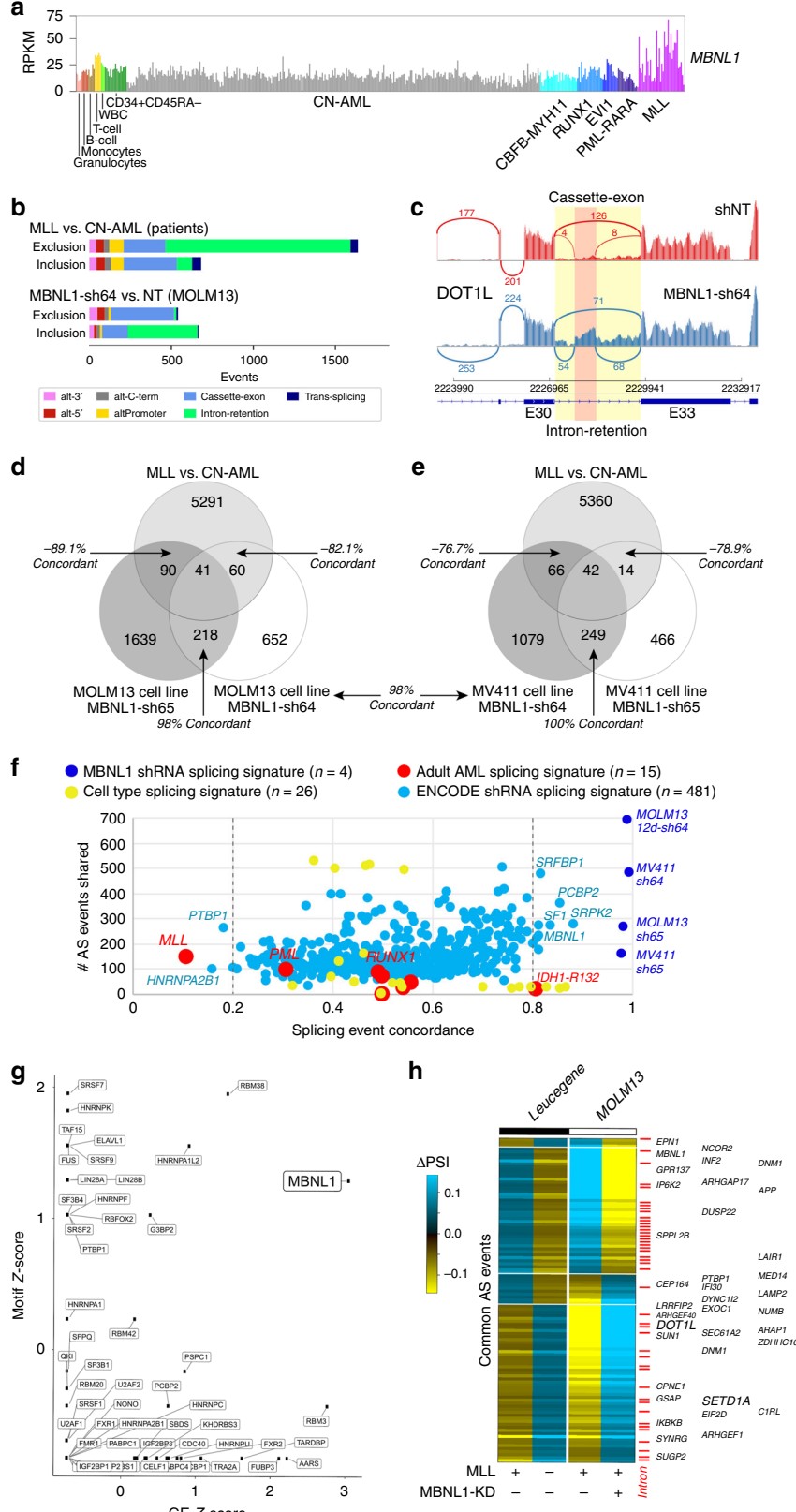

pathogenesis of MLL-rearranged leukemia[47,50] such as *DOT1L* and *SETD1A* may explain the growth inhibition that we observed. We first experimentally validated and characterized the splicing predictions identified by our analysis. Indeed, RT-PCR with primers spanning the predicted splice junctions confirmed the predicted AS events from our RNA-seq analysis (Fig. 7a). We also

performed RNA-immunoprecipitation using MBNL1-specific antibodies, and found substantial enrichment of *DOT1L* and *SETD1A* transcripts in precipitated RNA compared to control IgG (Fig. 7b), suggesting that MBNL1 interacts directly with the differentially spliced transcripts that we identified. These splicing changes lead to protein-level effects, with either decreases in total

**Fig. 6 Partial reversal of impaired intron retention in MLL-rearranged cells by MBNL1 knockdown. a** *MBNL1* gene expression levels in adult patient AML samples according to genetically/cytogenetically defined subtypes and for sorted cell populations. **b** Breakdown of the type of alternative splicing events detected when comparing Leucegene AML diagnostic patient RNA-Seq samples with prior reported MLL oncofusions versus cytogenetically normal patients (CN-AML), or events obtained from MOLM13 AML cells bearing the MLL-*AF9* karyotype with transfection of a short-hairpin RNA for MBNL1 (sh64) versus shRNA control (NT). Splicing-events that demonstrate increased exon inclusion versus exon-exclusion are shown separately. **c** SashimiPlot illustrating increased intron retention and cassette splicing of the MLL interacting protein *DOT1L*, resulting in alternate, likely truncated isoform in MBNL1 knockdown cells (shRNA construct 64). Exon number definitions are based on AltAnalyze Ensembl 72 definitions. **d**, **e** Overlapping alternative splicing events from MLL oncofusion patient samples compared to those from two different MLL-rearranged cell lines (MOLM13 (**d**), MV4;11 (**e**)), using two separate short hairpin RNAs against *MBNL1* (sh64, sh65). Callouts highlight the number of splicing events occurring anti-concordantly between the Leucegene MLL vs. CN-AML and shMBNL1 vs shNT comparison, or AS events occurring in the same direction (shMBNL1-64 and -65 overlap). Concordance percentages are indicated next to each overlap. Concordance between MV4;11 (sh64) and MOLM13 (sh65) is indicated between the two Venn diagrams. **f** Unbiased comparison of splicing signatures associated with different AML genetic/cytogenetic subtypes, prior splicing factor knockdown signatures and hematopoietic cell types relative to that of *MBNL1* knockdown. Splicing signatures were derived using the same pipeline for determining *MBNL1* knockdown and MLL rearrangement-associated splicing events. Similarity to the *MBNL1* knockdown splicing signature (specifically shMBNL1-64 in MOLM13) is shown in terms of splicing concordance, similar to (**d**) and (**e**), indicating the number of splicing events common to each signature on the Y-axis. Fifty-percent concordance is representative of random overlap. For each group, n represents number of pairwise comparisons (i.e., AML with fusion/molecular alteration vs CN-AML for Leucegene, RBP knockdown vs non-targeting control for ENCODE, hematopoietic cell type vs stem/progenitor, or shMBNL1 vs non-targeting control. **g** Statistical enrichment (z-score) of defined RNA-binding protein RNA-recognition elements (motifs) from the CisBP-RNA database for alternatively splicing exonic and intronic intervals with MBNL1 knockdown (sh64) compared to enrichment of differential gene expression (z-score). **h** Heatmap of all overlapping alternative splicing events in Leucegene MLL versus CN-AML and those observed with knockdown of MBNL1 (sh64) in MOLM13 cells, with callout of genes with prior implicated cancer relevance (ToppGene database). Red ticks indicates intron-retention events predicted from MultiPath-PSI.

protein or shifts in protein isoforms of DOT1L and SETD1A after *MBNL1* knockdown (Fig. 7c) likely due predominantly to nonsense-mediated decay arising from the incorporation of premature termination codons within retained introns. We subsequently performed RNA-seq on MBNL1 knockdown MOLM13 cells at a later time point, reasoning that splicing changes and alterations in histone methyltransferases may not immediately be reflected in gene expression if assayed at too early a timepoint. We managed to collect sufficient RNA for analysis at 12 days after knockdown, and found a marked increase in the number of differentially expressed genes when compared to the 4 day timepoint (Fig. 7d; Supplementary Data 3). Through gene set enrichment analysis (GSEA)[51] of our day 12 knockdown signature, we found significant corresponding bidirectional enrichment for gene sets associated with perturbation or disruption of central mediators of the MLL leukemia program, specifically HOXA9[52] and glycogen synthase kinase 3, previously shown to be required for MLL leukemogenesis by Wang et al.[53,54] (Fig. 7e). Together, these results suggest that loss of MBNL1 in MLL rearranged leukemia results in a state analogous to disruption of essential components of the MLL leukemia program, culminating in leukemia cell death.

## Discussion

Our findings described above demonstrate that MBNL1 is essential for MLL-rearranged leukemia cell growth, yet only modestly so for normal hematopoiesis. We also demonstrate using novel bioinformatic analyses that MLL-rearranged leukemias exhibit a unique pattern of mRNA isoform utilization that is partially reversed upon knockdown of *MBNL1*, suggesting that MBNL1-mediated alternative mRNA splicing patterns contribute to the pathogenesis of MLL-rearranged leukemias.

MBNL1 is well known to regulate RNA alternative splicing, localization, and decay[7–11], though an understanding of its role in disease pathogenesis has been largely limited to DM1 and Fuchs corneal endothelial dystrophy[42,55]. Recently, MBNL proteins have been reported to control the pluripotency of embryonic stem cells and fetal terminal erythropoiesis as well[13,14], implicating their activity to some extent in the hematopoietic system. In examining results of gene expression profiling of MLL leukemias, we identified *MBNL1* as one of the most consistently overexpressed genes

in this family of leukemia regardless of lineage, even more so than the *HOXA* cluster genes. This may be due to the fact that even within MLL-rearranged leukemias *HOXA* gene expression can be highly variable, especially in pre-B ALL[6,56,57]. Because we sought commonalities in gene overexpression across both myeloblastic and lymphoblastic leukemias, this may explain why the *HOXA* cluster genes were not consistently overexpressed in every dataset we examined.

The results of our RNA-seq and splicing analyses suggest that a major function of MBNL1 in MLL-rearranged leukemia is through alternative splicing, and a number of alternatively spliced genes identified using our approach have been implicated in leukemogenesis, including a subset of epigenetic regulators such as the H3K79 methyltransferase *DOT1L* and the SET1 histone methyltransferase complex member and cyclin K regulator *SETD1A*. A frequent splicing outcome following *MBNL1* knockdown was intron retention, and *MBNL1* knockdown appears generally to reverse intron exclusion. Our findings suggest that MLL-fusion proteins, via *MBNL1* overexpression, promote the expression of protein coding genes typically suppressed through intron retention-introduced premature termination codons. The negative impacts of *MBNL1* knockdown on cell growth are likely a net effect of multiple transcripts (and the subsequent proteins) being affected, rather than solely arising from alternative splicing of the products of a single gene. Ultimately, it remains challenging to quantify the exact extent to which MBNL1 or other splicing factors mediate specific oncogenic splicing programs in vivo, due to a lack of controlled model systems to quantify such effects. In our analysis, we compared *MLL*-rearranged patient samples to CN-AML as a control, however even within this comparison there are likely additional splicing differences attributable in part to phenomena such as lineage skewing[58], co-occurrence of spliceosome mutations within the patient cohort analyzed above[44], or other yet-unidentified *MLL* fusion-specific transcriptional/splicing programs, hence isolation of a core splicing factor-associated program in vitro will only remain an approximation of molecular programs found in patients. Despite this, we demonstrate a high degree of splicing discordance between MBNL1 loss and MLL fusion-associated patient splicing, indicating a high specificity of our analysis for MBNL1-mediated splicing events.

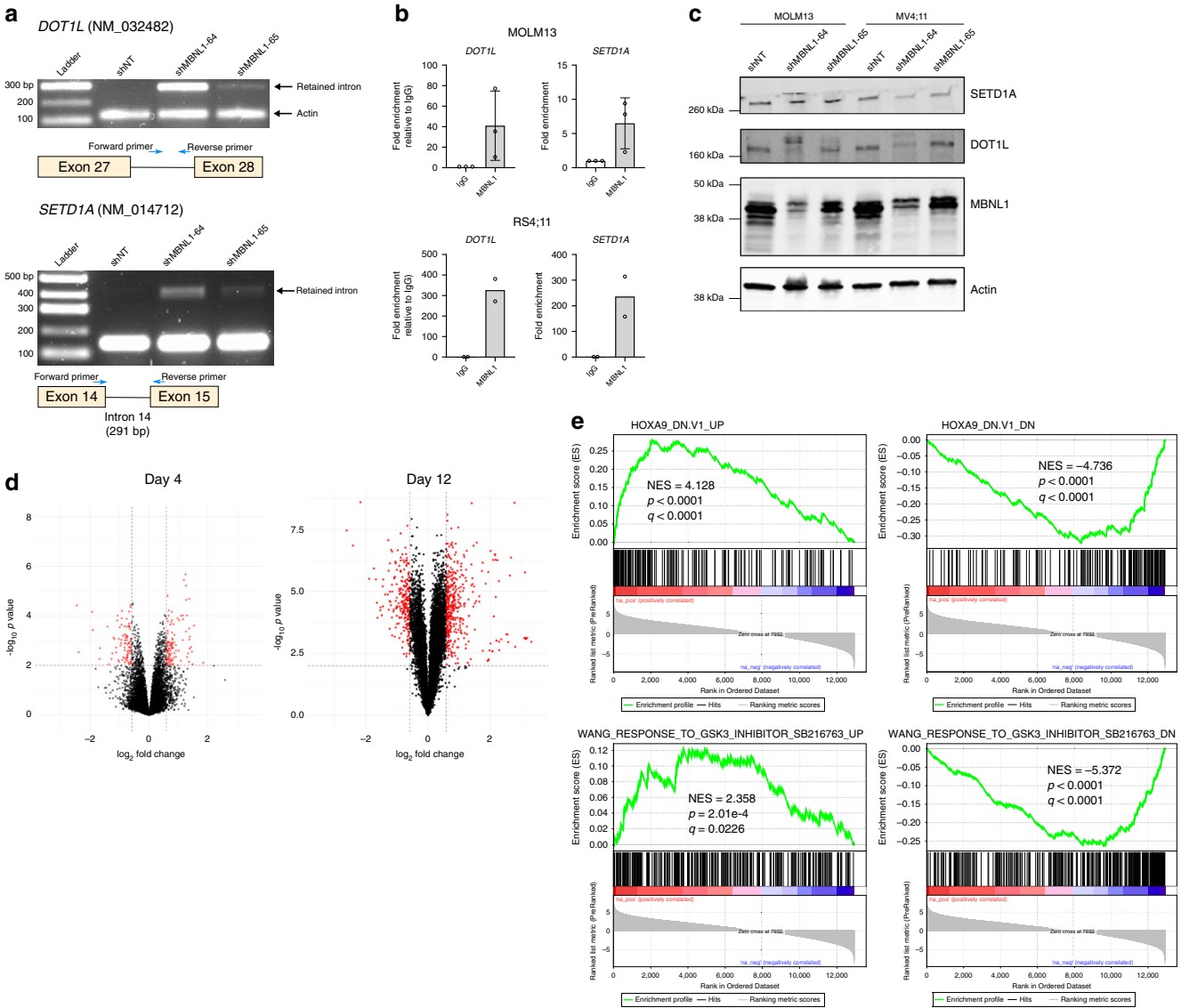

**Fig. 7 MBNL1 loss perturbs essential MLL-rearranged leukemia genes through missplicing. a** RT-PCR demonstrating the predicted increase in intron retention at alternative splice sites in *DOT1L* and *SETD1A* transcripts in MOLM13 cells after *MBNL1* knockdown. For *DOT1L*, *Actin* serves as a loading control, and primer product ("Intron Inclusion Band") only occurs when the intron is retained. Representative gel images shown, two biological replicates performed. **b** qRT-PCR results from RNA-immunoprecipitation assay assessing for enrichment of *DOT1L* and *SETD1A* transcripts in MBNL1-precipitated RNA. Error bars show mean ± SD. Data is from three biological replicates for MOLM-13 and two biological replicates for RS4;11. **c** Western blots of MOLM13 and MV4;11 cells following knockdown of MBNL1, demonstrating changes in DOT1L and SETD1A protein levels. Representative western blots shown, two biological replicates performed. SETD1A image represents same samples run and processed in parallel on separate blot. **d** Volcano plots representing differentially expressed genes in MOLM13 cells at 4 days (left) and 12 days (right) after knockdown. *X*-axis thresholds represent 1.5-fold change in expression; *Y*-axis threshold represents *p* < 0.01. Differentially expressed genes determined by empirical Bayes two-tailed moderated *t*-test. **e** Differentially expressed genes were significantly enriched positively and negatively by GSEA for corresponding gene sets related to perturbation of essential MLL leukemogenesis processes (i.e., HOXA9, upper GSEA plots; GSK3, lower plots.) Enrichment scores, *p* values, and false-discovery rate *q* values calculated as previously published[51].

All three members of the MBNL family regulate alternative splicing during development. *Mbnl1*$^{-/-}$ mice display splicing defects in heart and skeletal muscle yet do not present clinical features of DM1. Notably, *Mbnl2* was found to compensate for *Mbnl1* in murine tissues[59]. We speculate that a similar phenomenon might explain the eventual development of leukemia in our *Mbnl1*$^{-/-}$ mice despite the growth inhibition in our knockdown studies in human cells. In both *Mbnl1*$^{+/+}$ and *Mbnl1*$^{-/-}$ states *Mbnl2* and *Mbnl3* are stably expressed in mouse leukemia cells, however neither analogue was meaningfully expressed in the human leukemia lines assayed (Supplementary Fig. 7A–B).

Therefore, one possible explanation for the eventual development of leukemia in our *Mbnl1*$^{-/-}$ murine model is that *Mbnl2* compensates for the absence of *Mbnl1*, but it is unclear if this compensatory mechanism would necessarily behave similarly in human cells. Alternately, it is plausible that MBNL1 loss in a transformed cell (i.e., genetic knockdown in leukemia cell lines) may be more deleterious than de novo transformation of an MBNL1-deficient cell, as occurs in the mouse retroviral leukemia model. We hypothesize that transformation of *Mbnl1*$^{-/-}$ cells could conceivably engage alternate oncogenic pathways which eventually allow leukemia cell survival, though this issue of

MBNL1 requirement for MLL leukemia initiation versus propagation requires further study.

Given the poor prognosis of patients with MLL-rearranged acute leukemia, there remains an urgent need for better therapeutic approaches in this subgroup of patients. Our data suggests that targeting MBNL1 may be a valid therapeutic avenue in the treatment of MLL-rearranged leukemia. As a proof of concept, we demonstrate that a small-molecule inhibitor rationally designed against MBNL1 can preferentially kill MLL-rearranged leukemia cells while sparing normal CD34+ hematopoietic stem/progenitor cells. An alternative therapeutic approach may involve targeting elements of the *MBNL1* transcript itself using therapeutic RNA inhibition specifically targeted at hematopoietic or leukemic cells. Specifically, our data suggest that human MLL-rearranged leukemia cells have increased expression of *MBNL1* transcripts that exclude exon 5 which influences nuclear vs cytoplasmic localization of MBNL1[14,43]. We hypothesize that the role of MBNL1 in facilitating productive splicing of the MLL-rearranged oncogenic program may in part be mediated by exon 5 exclusion. Therapeutics which influence exon utilization (such as antisense oligonucleotides) may thus represent an alternate means of disrupting the function of MBNL1. However, the specific role of the exclusion transcript and the associated cellular localization are not fully understood and require further study.

In conclusion, we provide evidence supporting a model in which *MBNL1* overexpression mediated by the MLL-fusion protein promotes leukemogenesis via an altered splicing profile (primarily changes in intron retention) of MBNL1 targets. We demonstrate that MBNL1 is an important regulator of alternative splicing events preferentially expressed in MLL-rearranged leukemia and which participate in the MLL-rearranged oncogenic program. Our data shows the effect of MBNL1 at the mRNA and protein level, and suggests that MBNL1 plays a key role in the pathogenesis of MLL-rearranged leukemia by stabilizing the transcripts of multiple leukemogenic genes including *DOT1L* and *SETD1A*; this in turn supports transcriptional activation of downstream targets of the MLL-fusion protein, including activation of MBNL1, creating a positive feedback loop (Fig. 8). We also show that a specific, albeit low potency, inhibitor of MBNL1 preferentially affects MLL-rearranged leukemia cells. Given recent discoveries supporting the concept of selective RNA binding protein essentiality in AML[60], our work provides further evidence that this is a valid new class of targets in this poor-prognosis class

of diseases. Furthermore, our work serves as proof of principle for development of anti-MBNL1 therapies. Additional research is ongoing to further elucidate the mechanisms underlying MBNL1 dependence in MLL-rearranged leukemia, as well as to further optimize candidate small-molecule inhibitors.

## Methods

**Cell lines and primary patient cells/data**. Human leukemia cell lines were maintained in Iscove's modified Dulbecco medium (IMDM) or Roswell Park Memorial Institute (RPMI) 1640 medium supplemented with 10% fetal bovine serum (FBS), 1% penicillin, and 1% streptomycin.

Fully deidentified primary patient cells were obtained from the Cincinnati Children's Hospital Medical Center Biorepository. Patient identities have been protected. PDX models were generated using residual diagnostic specimens according to an IRB approved protocol (#2008-0021) following proper informed consent. Cells from patient samples were transplanted into busulfan-conditioned NOD/LtSz-SCID interleukin-2(IL2)RG−/− SGM3 (NSGS) (Jackson Laboratories, stock no. 013062) mice. By the time the mice displayed signs of leukemia, the only human cells that remained in the mice were leukemic cells (MLL-rearrangement was confirmed with FISH). De-identified PDX spleen samples taken from NSGS mice (Jackson Laboratories, stock no. 013062) were used for the ex vivo knockdown experiments described below. Cells were cultured in IMDM supplemented with 20% FBS and 10 ng/ml human cytokines including SCF, FLT3-Ligand, Thrombopoietin, IL-3, and IL-6.

MLL-AF9 Tet-off human CD34+ cells were a kind gift from Dr. James Mulloy, and were generated as follows. Umbilical cord blood (UCB) was obtained from the Translational Trial Development and Support Laboratory (TTDSL) of CCHMC. UCB units are collected and distributed by the TTDSL under an IRB-approved protocol, and originate from discarded umbilical cords and are thus completely anonymized, rendering them exempt from human subjects research. CD34+ cells were isolated from UCB using the EasySep CD34+ isolation kit (StemCell Technologies). CD34+ cells were pre-stimulated in IMDM/20%FBS containing 100 ng/mL SCF, TPO, Flt3-L, and IL-6 and 20 ng/mL IL-3 for 2 days before transduction. Transduction was carried out using spinoculation onto Retronectin (Takara Bio) coated plates along with 4 μg/mL polybrene. Cells were trasduced with both the pSIN-TREtight-dsRED-MLL/AF9[35] lentivirus and the MSCV-GFP-IRES-tTA[34] retrovirus. After transduction, cells were maintained in IMDM/20% FBS with 10 ng/mL of each cytokine. A pure population of dsRED+GFP+ cells were selected over several weeks of culture. To repress expression of the MLL fusion, cells were treated with 1 μg/mL doxycycline.

Cell lines were periodically validated by STR genotyping through Genetica Cell Line Testing (LabCorp). Cells were tested and were negative for mycoplasma contamination. None of the cell lines utilized in this study are recognized by the ICLAC as being commonly misidentified.

RNA-Seq data for primary patient AML and ALL samples were obtained from the St. Jude Cloud (originating publications/datasets as cited in the "Results" section), and from www.vizome.org/aml as indicated.

**Mouse experiments**. All animals used for this study were 6–12-weeks old. All animal experiments were carried out in accordance with the guidelines of the Institutional Animal Care and Use Committee (IACUC). C57BL/6 mice were used as donors for transplantation experiments using $Mbnl1^{-/-}$ or $Mbnl1^{+/+}$ cells. For studies of normal murine hematopoiesis, $1 \times 10^6$ CD45.2 Mbnl1$^{-/-}$ or Mbnl1$^{+/+}$ bone marrow donor cells were transplanted with $1 \times 10^6$ CD45.1 competitor cells (for competitive transplantation, 1:1 ratio) or without competitor cells (non-competitive transplantation) into lethally irradiated BoyJ recipient mice (RRID: IMSR_JAX:002014). Four or eight weeks after transplantation, bone marrow of recipients was analyzed for donor chimerism. Mice were assessed for signs of leukemia, and when distressed, were sacrificed and tested for leukocytosis, anemia, thrombocytopenia and splenomegaly. For normal hematopoiesis experiments, peripheral blood of recipients was examined every 4 weeks and bone marrow was tested at the end point of the experiment for the presence of donor cells by flow cytometry. For studies of time-to-leukemia in $Mbnl1$ knockout mice, BoyJ recipient mice were conditioned by sub-lethal irradiation and i.v. injected with $2 \times 10^5$ transduced donor cells. For xenograft experiments with MLL-rearranged cell lines, immunocompromised NOD/LtSz-SCID interleukin-2(IL2)RG−/− (NSG) (Jackson Laboratories, stock no. 005557) recipient mice were treated with busulfan (Sigma) 30 μg/mg intraperitoneally and transplanted with $2 \times 10^5$ human cells 24 h later. For xenograft experiments with MLL-rearranged primary patient cells, immuno-compromised NOD/LtSz-SCID interleukin-2(IL2)RG−/− SGM3 (NSGS) (Jackson Laboratories, stock no. 013062) recipient mice were conditioned with busulfan and transplanted with $2–7.5 \times 10^5$ human cells 24 h later. In xenograft experiments bone marrow samples were collected four weeks after transplantation as well as when signs of leukemia were present; aspirates were analyzed via flow cytometry for the presence of human CD45+ cells and the presence of shRNA-transduced CD45+ and Venus-positive cells

**Retroviral and lentiviral transductions**. Retroviral and lentiviral supernatants were generated by transfection of HEK293T cells using the CaPO$_4$ method or

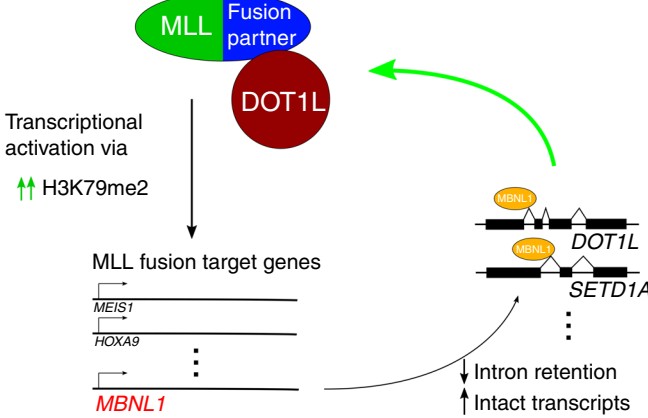

**Fig. 8 Working hypothesis for the role of MBNL1 in MLL-rearranged leukemia.** The MLL-fusion protein directly promotes MBNL1 gene overexpression, resulting in changes in alternative splicing of target RNA transcripts (typically decreased intron retention and therefore decreased nonsense-mediated decay/increased intact transcripts) which are central to the MLL-rearranged leukemia program.

FuGENE 6 reagent (Promega) according to the manufacturer's recommendations. Mouse bone marrow cells were isolated by crushing ischial bones, femurs and tibias. Cells were then lineage-depleted using the Lineage Cell Depletion kit (Miltenyi Biotec) following magnetic separation by using an AutoMACS Pro Separator (Miltenyi Biotec). Lineage-negative cells were then pre-stimulated with 100 ng/ml cytokines (SCF, G-CSF, and Thrombopoietin) (Peprotech) overnight and transduced with retroviral MLL-AF9 constructs using spinoculation on retronectin-coated plates.

The lentiviral MBNL1 shRNA pLKO.1-Puro plasmids (TRCN0000063963, TRCN0000063964, TRCN0000063965, TRCN0000063966, and TRCN0000063967) were purchased from Millipore Sigma. Human cells were incubated with a single dose of lentiviral supernatant. Cells transduced with constructs granting puromycin resistance were selected in 0.5–5 µg/ml of Puromycin for 72 h. Cells transduced with constructs containing a fluorescent marker (GFP, YFP or Venus) were isolated on day 4–5 after transduction by sorting using a MoFlo XDP (Beckman Coulter), FACSAria (BD Biosciences), or a Sony SH800S (Sony Biotechnology.) MSCV MLL-AF9 GFP was a kind gift from Dr. Gang Huang.

**Flow cytometry**. For murine peripheral blood flow cytometry analysis, peripheral blood was collected by retro-orbital bleeding and incubated with surface antibodies (eBioscience) in staining buffer (phosphate-buffered saline (PBS) supplemented with 0.5% FBS). Peripheral blood cell suspensions were treated with Pharm Lyse Lysing Buffer (BD Biosciences) and washed twice with PBS. For apoptosis assays, cells were incubated with allophycocyanin-conjugated Annexin V (BD Bioscience) for 15 min at RT in 1X Annexin V Binding Buffer (BD Bioscience) following staining with 7-aminoactinomycin (eBioscience). Data were acquired on a FACS-Canto I and results were analyzed using FlowJo Version 10 (FlowJo). The list of antibodies used for flow cytometry analysis of mouse and human cells is indexed in Supplementary Data 4.

**Colony-forming unit assays**. Mouse bone marrow cells were plated in methylcellulose-containing media (StemCell Technologies, Methocult M3434) supplemented with 10 ng/ml of granulocyte-macrophage colony-stimulating factor (Peprotech). After 7–8 days of culture, colonies were counted and re-plated for 3–5 rounds. Human cells were cultured in methylcellulose-containing media (StemCell Technologies, H4434). Colonies were scored 14 days after plating. The Nikon Eclipse Ti-U scope was used for images of colonies and automatic counting was performed using Nikon Elements software.

**RT- and quantitative RT-PCR**. Total RNA was extracted from human Puromycin-selected or sorted Venus-positive cells using the RNeasy Mini kit (Qiagen). RNA was reversed transcribed into cDNA using iScript Advanced cDNA Synthesis kit (Bio-Rad Laboratories). RT-PCR reactions were carried out in an Eppendorf PCR Cycler using MyTaq™ HS Red Mix (Bioline) according to the manufacturer recommendations. The amplified reaction products were separated on an agarose gel (ranging from 1 to 3.5%). The BioChemi System (UVP Bioimaging system) was used for visualization and analysis (Image Lab 6.0 used for band quantification). For quantitative RT-PCR 5–10 ng of cDNA was analyzed using iTaq Universal SYBR Green Supermix and iTaq Universal Probes Supermix (Bio-Rad) in a Ste-pOnePlus Real-Time PCR machine (Applied Biosystems). Primers for RT-PCR are detailed in Supplementary Data 5.

**RNA-Immunoprecipitation**. RNA-Immunoprecipitation was performed using the Magna RIP kit (Millipore). $2.0 \times 10^7$ cells were used per immunoprecipitation reaction. Five micrograms of either mouse anti-IgG (Millipore, CS200621) or anti-MBNL1 (Millipore, clone 4A8) was used. After RNA-immunoprecipitation, RNA was extracted and analyzed via qRT-PCR through the process described in the "RT- and Quantitative RT-PCR" section, with the exception that the qRT-PCR was run using Taqman Probes (Thermo Fisher Scientific, Hs_00986924 for *SETD1A* and Hs_05017433 for *DOT1L*) instead of SYBR Green.

**Western blotting**. The primary antibodies used were anti-MLL1 (Bethyl, A300-086A, RRID: AB_242510), anti-MBNL1 (Millipore Sigma, clone 4A8, RRID: AB_10808499), anti-DOT1L (Cell Signalling, D1W4Z, RRID:AB_2799889), anti-SETD1A (Cell Signalling, D3V9S, RRID:AB_2799614), anti-Actin (Cell Signalling Technology, 13E5, RRID: AB_2223172), anti-Lamin B1 (Cell Signaling Technology, D4Q4Z, RRID: AB_2650517) and anti- β-tubulin (Cell Signaling Technology, 9F3, RRID: AB_823664). Whole cell lysates were isolated using RIPA buffer (Sigma) while nuclear extracts were obtained using the NE-PER kit (Thermo Scientific) according to manufacturer's instructions, and the amount of protein was determined by the BCA Protein Assay Kit (Thermo Scientific). Ten µg of protein was separated by SDS-PAGE on a 4–20% gradient gel (Bio-Rad). After transfer to PVDF membranes, blots were blocked with Odyssey® Blocking Buffer TBS (LI-COR) for one hour and incubated with primary antibodies overnight. After washing, blots were treated with appropriate secondary IRDye 680RD goat anti-mouse (LI-COR) and IRDye 800CW goat anti-rabbit (LI-COR) antibodies at a dilution of 1:10,000 for one hour. Images were obtained using the Odyssey CLx Infrared Imaging System (LI-COR).

**Inhibitor assay**. $0.5 \times 10^5$ cells/ml were plated in a 96-well plate and incubated with the inhibitor at a concentration of 250 µM, 500 µM, or vehicle (DMSO). At various time points after incubation cells were stained with Annexin V and 7-AAD and analyzed by flow cytometry; additionally, viable cells were counted using Trypan Blue.

**ChIP-seq and microarray reanalysis**. For Fig. 1a, pre-computed microarray gene expression signatures for MLL-rearranged versus WT leukemias were obtained from the following sources: Ross et al., *Blood* 2004 Supplementary Table 10[4], Mulligan et al. *Leukemia* 2007 Suppl Table 5[22], Zangrando et al. *BMC Med Genomics* 2009 Suppl Table 2[21] and Stam et al. *Blood* 2010 Suppl Table 5[6]. Probe IDs were reconciled with gene symbols, and intersections were determined and visualized with http://bioinformatics.psb.ugent.be/webtools/Venn/.

For Fig. 1c, previously analyzed bigWig/bedgraph files were downloaded from the Gene Expression Omnibus using the listed accession numbers (GSE95511 for ML-2 data, GSE79899 for MV4;11 and THP-1 data, GSE38403 for RS4;11 data, and GSE38338 for SEM data) and visualized with the UCSC Genome Browser, assembly hg19. For consistency, data for RS4;11 and SEM were converted from their originally mapped hg18 to hg19 using the liftOver tool from the UCSC Genome Browser Utilities.

**Statistics**. All means were compared using two-tailed t-tests unless specifically indicated otherwise. For survival analyses, log-rank tests were used to compare survival curves. The statistics used for the RNA sequencing analyses are described in the section below ("RNA sequencing and analysis"). Graphpad Prism 8 was used to perform statistical tests and create graphs. Select figures created with BioRender.com.

**RNA sequencing and analysis**. RNA sequencing was performed on total RNA obtained from MOLM13 or MV411 cell lines following transduction with either MBNL1 sh64 (n = 3 MOLM13, n = 3 MV411), sh65 (n = 4 MOLM13, n = 4 MV411) or shNT control (n = 3 MOLM13, n = 4 MV411) at a depth of >40 million paired-end stranded reads using the Illumina HiSeq 2500 (GSE123441), following library preparation with the TruSeq RNA Library Prep Kit (Illumina), similar to that of the Leucegene RNA-Seq dataset (GSE67040). A longer 12-day MBNL1 knockdown with independent shNT controls (MOLM13) was also performed as biological triplicates, processed with the same protocol on an Illumina NovaSeq platform. AML FASTQ files for MLL (n = 27) and CN AML patients (n = 29) were downloaded from the GEO database as previously described[44]. All FASTQ files were aligned to the reference human genome (GRCh37/hg19) using the latest version of STAR[61]. Exon-exon and exon-intron spanning reads were determined from the software AltAnalyze (version 2.1.1) using the Ensembl-72 human database, along with splicing event calculation and annotation with the MultiPath-PSI algorithm (see: http://altanalyze.readthedocs.io/en/latest/Algorithms for algorithm details and benchmarking). For comparative splicing analyses, >50% of samples for each splicing event were required to have detected PSI value, consistent between the patient dataset and transduced cell line analyses. As statistical thresholds, we used a two-tailed empirical Bayes moderated (eBayes) t-test with p ≤ 0.01 for shRNA analyses and delta PSI > 0.1. Breakdown of splicing event categories is provided through the MultiPath-PSI software and associated heatmap visualization through AltAnalyze. For the MLL patient samples versus CN, an eBayes t-test was also performed (p ≤ 0.01) and delta PSI > 0.1 was used for differential splicing analysis. Only unique junction-cluster ID results were reported. Relative PSI differences were calculated separately for patient dataset and transduced cell line samples for joint visualization. Gene expression analyses were similarly performed in AltAnalyze from the input STAR BAM files, using the same p-value thresholds and 1.5 fold change cutoff, with gene-level RPKM quantification. ENCODE RNA binding protein knockdown and control hematopoetic cell type splicing comparison results were used from a prior study as previously described[62]. RNA recognition element (RRE) enrichment analysis was performed on alternatively spliced exons and introns (exons with 500nt flanking intron sequence bins) using the software HOMER[63] from a large database of RRE motif models in the form of position frequency matrices (PFMs) taken from the CisBP-RNA database[64]. Evaluation of spilcing concordance was performed using a custom Python script. Pathway enrichment analysis was performed with ToppGene[65]. GSEAPreranked[51] was performed on differentially expressed genes as determined by AltAnalyze.

**Reporting summary**. Further information on research design is available in the Nature Research Reporting Summary linked to this article.

## Data availability
The RNA-seq data (MOLM13 and MV4;11 cells with shNT versus multiple shMBNL1 knockdown conditions) have been deposited to the NCBI Gene Expression Omnibus (GEO) under accession number GSE123441.

The RNA-seq data for pediatric leukemia samples used for analysis in this study were obtained from the hematologic malignancies patient cohort of the St. Jude Cloud (https://pecan.stjude.cloud), a publicly accessible pediatric genomic data resource requiring approval for controlled data access.

The RNA-seq expression data from the 'BeatAML' dataset was obtained from www.vizome.org/aml, a publicly accessible AML genomic data resource, using the gene

expression viewer stratified by WHO fusion category. Original data used to generate Fig. 6b, d, e, f, h are included as Supplementary Tables 1 and 2. The original data to generate Fig. 7d is included as Supplementary Table 3. The original gels and blots for Figs. 1d, 2a, 5, 7a and c are included as Supplementary Data. All the other data supporting the findings of this study are available within the article and its supplementary information files and from the corresponding author upon reasonable request.

## Code availability

AltAnalyze is an open source software package, and freely available for download at www.altanalyze.org

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

## Acknowledgements

We would like to thank Maurice Swanson, PhD, for providing the *Mbnl1*−/− transgenic mice. We are also grateful to Matthew Disney, PhD, for providing the small molecule MBNL1 inhibitor. The SH800S is supported by an NIH Shared Instrumentation Grant (S10OD023410). A.G. was supported by a CancerFree KIDS grant. N.S. was supported by an NIH grant (R01 CA226802). M.W. was supported by an NIH grant (R50 CA211404). A.R.K. was supported by a Hyundai Hope on Wheels grant. L.H.L. was supported by a CancerFree KIDS grant as well as an NIH grant (L40 HL143713-01). L.H.L. is a St. Baldrick's Foundation Scholar.

## Author contributions

S.S.I., A.R.K., A.G. and L.H.L. conceived and designed the experiments. A.G., S.S.I., J.C., M.B., M.W., M.R.B. and L.H.L. performed experiments and analyzed experimental data. N.S. designed and oversaw bioinformatics analyses; N.S., K.C., A.K., and M.V. performed the informatics analyses. All authors contributed to the writing and editing of the manuscript.

## Competing interests

The authors declare no competing interests.
