## [Peer Review File · Nature Communications]

Reviewers' comments:

Reviewer #1; Alternative splicing

This is an interesting manuscript which suggests that the RNA binding protein MBNL1 is required for leukemogenesis of MLL-rearranged leukemia and serves as a direct binding target of the MLL-fusion oncoprotein. Several key issues should be dealt with to clarify the claims in the manuscript as follows:

-The analyses of the effects of MBNL1 on splicing are confusingly presented and not clear. First, an unbiased analysis of the effects of MBNL1 loss on splicing and gene expression using >1 shRNA and in >1 AML cell line are needed. The analyses in Figures 5-6 depend on a single shRNA and/or comparisons which are not necessarily helpful or correct. For example, it is not clear why comparison of splicing changes in MBNL1 loss in AML cells versus fibroblasts is helpful—it is quite possible that the effects of MBNL1 loss would be different across these cell types. In addition, comparison of splicing in MLL-rearranged versus non MLL-rearranged AML in the primary analysis of MBNL1 effects on splicing is not necessarily helpful as it is not clear from the data presented that MBNL1 upregulation is specific to MLL-rearranged AML.

-Related to the above, the effects of MBNL1 loss on each category of splicing (cassette exons, intron retention, 5' splice site, 3' splice sites, etc) should be analyzed. If intron retention is seen with MBNL1 loss then it is quite possible that MBNL1 is required for cassette exon inclusion as well and there may be an even greater effect of MBNL1 on cassette exon splicing than on intron retention.

-The concept that MBNL1 loss impact splicing but not gene expression does not make sense. Ultimately, MBNL1 loss must impact gene expression given its impact on MLL-rearranged AML cells shown. Evaluation of the effects of MBNL1 on gene expression at additional timepoints may be needed to discern the effects of MBNL1 loss on gene expression.

-The experiments using the MBNL1 inhibitor are quite preliminary. There is no evaluation of this compound on normal hematopoietic cells, it is not used in vivo (it is not clear if that is even feasible with this compound), the evaluation of this compound on non-MLL rearranged AML is minimal, and there is no evaluation of the effect of this compound on splicing or gene expression.

-Related to the above, it is not entirely clear if MBNL1 is required in only MLL-rearranged AML as claimed in the manuscript. Evaluation of genetic requirement for MBNL1 in non-MLL rearranged AML should be performed across a greater number of AML cell lines (only a single non-MLL rearranged cell line is tested in Figure 2). There are also public CRISPR data across AML cell lines which could be useful in this regard.

-Is Mbnl1 expression increased in the mouse MLL-AF9 retroviral model? It is very surprising that HOXA genes are not found as universally upregulated in MLL-rearranged leukemias in Figure 1. Can the authors comment on this?

-The "control" for MBNL1 expression in Figure 1B is not necessarily appropriate. How does the expression of MBNL1 in AML primary patient samples compare to normal adult CD34+ cells?

-It appears from Figure 4G that Mbnl1 is required for self renewal of normal mouse hematopoietic stem cells. What was the chimerism of HSPCs in primary transplant? Was a secondary competitive transplant attempted?

-Targeting of RNA binding proteins is not a "new paradigm in targeted therapy." This phrasing in the Abstract should be revised for accuracy. There have been numerous prior and ongoing efforts to therapeutically target RNA binding proteins in a variety of disorders (including very recent papers on targeting RNA binding proteins in AML).

Reviewer #2; computational splicing

This paper studies the effects of MBNL1 on aberrant alternative splicing in MLL-rearranged leukemia. Overall, this is a well designed and well executed study, and the results should be of broad interests to readers of Nature Communications. However, I have several concerns about the

analysis and interpretation of RNA-seq data, that need to be addressed in a revised manuscript.

1. I am surprised that the overlap of differential alternative splicing events found in patient samples and in MBNL1 knockdown cells is low (only 85 events at the intersection of the two sets – Fig. 5D). Is this degree of overlap no different from or significantly higher than what would be expected by random chance? Do the authors have any explanation for why the overlap is low?

2. As related to the point above, the authors should perform motif scans of RBP/splicing factor motifs, to see if the motifs for certain splicing factors are enriched around the differential alternative splicing events found in patient samples, or in MBNL1 knockdown cells, or in both. This may help answer why a large number of alternative splicing events are not shared between these two systems.

3. It is also somewhat surprising that intron retention is the most frequent type of alternative splicing events among the overlapped events, given that as a well-defined splicing factor MBNL1 should regulate many exon skipping events. I notice that the authors used AltAnalyze for the RNA-seq splicing analysis. Could this result be due to some inherent features of AltAnalyze? I think the authors should re-analyze their data using another popular software (for example, rMATS) to see if the breakdown between different types of alternative splicing events still holds.

Reviewer #3; MLL models

The manuscript by Gurunathan et al describe the identification of MBNL1 as an essential regulator of alternative splicing in MLL-rearranged (MLLr) AML. Specifically, the authors describe the selective up regulation of MBNL1 in MLL-rearranged AML and provide some evidence that MLL-fusion proteins directly regulate MBNL1. Down regulation of MBNL1 in MLLr cell lines inhibits growth both in cell lines and in vivo using primary patient material. On the molecular side, the authors identify interesting MBNL1 targets such as DOT1L and SET1D1A in both patient material and cell lines. Specifically, down regulation of MBNL1 promotes intron inclusion in DOT1L and SET1D1A presumably leading to NMD-sensitive transcripts and subsequent reduction of the cognate proteins. Finally, the authors provide some evidence for the potential of targeting MLLr using small molecules directed against MBNL1. Overall, the authors present a number of interesting findings. Experiments appear well executed and the data largely supports the conclusions drawn.

Major points

1) Overall the bioinformatics is not well described. A) In the analysis presented in Fig 1A, what does the authors mean by “up regulated in MLLr AML/ALL”. Is this relative to other leukemic subtypes or relative to normal cells (if so which)? I failed to find any description of how this analysis was performed. B) For the RNA-seq the authors identify commonly altered splicing events between patient (MLLr vs. other AMLs) and cell line (+/- KD of MBNL1). However, they don't describe what the frequency of these common events are (i.e. how many are common out of a total number of de-regulated events). This is especially important as they argue for the use of RNA splicing as a better predictor than gene expression. I suggest the authors make a Venn diagram for the splicing data similar to that in Figure 5D.

2) The authors argue for a selective effect of MBNL1 inhibition in normal/non-MLLr versus MLLr cells. This is supported by cell growth and survival experiments in a human setting. Overall, I think the authors are over-interpreting the MLLr specificity of MBNL1.

For cell line experiments, an important determinant is the growth rates and it is clear that the Kasumi-1 cell in Fig 2 are growing very slowly. To what extent is this true for the cells used in the inhibitor experiments, i.e. what are the relative growth rates of the cells used. Also, why are K562

and HL60 used for the inhibitor and Kasumi1 for the shRNA KD experiments?

3) The authors use an MBNL1 KO line to assess the in normal hematopoiesis and find relative mild effects. However, they do note a marked reduction in engraftment in a competitive transplantation setting suggestive of a stem cell defect. This could be tested by limited dilution experiments and/or by FACS-based quantification of stem cell numbers.

The authors further argue that the relative mild delay in leukemic development in MBNL1 KO mice could be due to the presence of MBNL2. This could be tested by performing shRNA KD experiments in this setting. The authors may also test the specific requirement for MBNL1 in MLLr leukemias by conducting similar experiments using other leukemic fusion proteins. This would strengthen the argument for the specificity of MBNL1 for MLLr leukemia.

Minor point

1) The authors use MLL-AF9 Tet-off human CD34+ cells but do not provide a description of their generation or origin. This should be included.

We wish to thank our reviewers for your constructive criticism and thoughtful feedback regarding our manuscript “MBNL1 regulates essential alternative RNA splicing patterns in MLL-rearranged leukemia” (NCOMMS-19-02864.) Please find attached a revised version of our manuscript with additional experimental data addressing the critiques raised by our reviewers. Our responses to their specific critiques are appended below.

Reviewer #1; Alternative splicing

The analyses of the effects of MBNL1 on splicing are confusingly presented and not clear. First, an unbiased analysis of the effects of MBNL1 loss on splicing and gene expression using >1 shRNA and in >1 AML cell line are needed. The analyses in Figures 5-6 depend on a single shRNA and/or comparisons which are not necessarily helpful or correct. For example, it is not clear why comparison of splicing changes in MBNL1 loss in AML cells versus fibroblasts is helpful—it is quite possible that the effects of MBNL1 loss would be different across these cell types. In addition, comparison of splicing in MLL-rearranged versus non MLL-rearranged AML in the primary analysis of MBNL1 effects on splicing is not necessarily helpful as it is not clear from the data presented that MBNL1 upregulation is specific to MLL-rearranged AML.

R1 response 1. We appreciate this reviewer’s point. We have completely rewritten our analysis, and subsequently performed additional RNA-seq and alternative splicing analysis using a second shRNA against *MBNL1* in MOLM-13 cells, in addition to a new series of experiments (control vs two separate *MBNL1* shRNAs) in a second AML cell line (MV4;11) which carries the *MLL-AF4* translocation. These analyses indicate a high concordance in splicing changes between hairpins and/or MOLM-13 and MV411 and the specificity of these changes for MLL rearrangements (**Figure 6D-F; line 300**). We also clarify our rationale for initially examining splicing of targets identified in fibroblasts: while MBNL1 likely regulates the splicing of genes involved in entirely unique processes in AML, we show that there are still common MBNL1-mediated splicing changes which occur upon knockdown/inhibition regardless of tissue type or disease state, even if those genes are not necessarily implicated in AML pathogenesis. We subsequently utilize these as indicators of on-target inhibition of MBNL1 by our academic compound, even though they are unlikely to play a major role in the observed phenotype (**Figure 5A-C; line 272**). We further now show the extent and variability of *MBNL1* upregulation among different cytogenetic subtypes of AML, which indicates specific upregulation of MBNL1 with MLL translocations (**Figure 6A**).

Related to the above, the effects of MBNL1 loss on each category of splicing (cassette exons, intron retention, 5’ splice site, 3’ splice sites, etc) should be analyzed. If intron retention is seen with MBNL1 loss then it is quite possible that MBNL1 is required for cassette exon inclusion as well and there may be an even greater effect of MBNL1 on cassette exon splicing than on intron retention.

R1 response 2. Based on the reviewer’s recommendations, we now enumerate the absolute proportion and direction of regulation for each category of alternative splicing/transcription. CN-AML and *MBNL1* knockdown in an MLL-AF9 cell line (**Figure 6B**). Here, intron retention represents the principal impact of *MBNL1* knockdown (increased with knockdown) along with a global decrease in exon inclusion. In adult AML patients with MLL rearrangements, the principal splicing difference is a decrease in intron retention compared to CN-AML patients. For reference, this extent of intron retention region is indeed high when compared broadly to other subtypes of adult AML with genetically defined splicing

dysfunction (*U2AF1*-S34X, *SRSF2*-P95X, *SF3B1*-K700E, *ZRSR2*, etc.) (**Figure R1A, Supplemental Figure S5A**). Notably, *ZRSR2* mutations, but not other predominant AML splicing factor mutations (*U2AF1*, *SRSF2*, *SF3B1*), are well established to result principally in intron retention in AML and MDS (Thol et al. Blood, 2012). MLL rearrangements results in a similar but opposite magnitude of intron retention to *ZRSR2*. As to the biological impact, intron retention principally results in increased nonsense mediated decay in through the introduction of premature stop codons, which would decrease overall protein abundance with *MBNL1*-knockdown as demonstrated specifically here for *DOT1L* and *SETD1A*.

The concept that MBNL1 loss impact splicing but not gene expression does not make sense. Ultimately, MBNL1 loss must impact gene expression given its impact on MLL-rearranged AML cells shown. Evaluation of the effects of MBNL1 on gene expression at additional timepoints may be needed to discern the effects of MBNL1 loss on gene expression.

R1 response 4. We agree with the reviewer's point. We subsequently performed a longer-term culture of MOLM-13 cells cultured out to 12 days post-transduction under selective conditions. At this stage, we observe a proportionally greater number of differentially expressed genes when analyzed using AltAnalyze. Furthermore, upon performing functional enrichment analyses with both ToppGene and GSEA, we note significant enrichment for perturbation signatures (both up- and down-regulated) arising from interference with essential MLL-fusion leukemia pathways, which we believe further supports our hypothesis that *MBNL1* regulates common MLL fusion (**Figure 7D,E; line 366**).

The experiments using the MBNL1 inhibitor are quite preliminary. There is no evaluation of this compound on normal hematopoietic cells, it is not used in vivo (it is not clear if that is even feasible with this compound), the evaluation of this compound on non-MLL rearranged AML is minimal, and there is no evaluation of the effect of this compound on splicing or gene expression.

R1 response 5. We did not have sufficient compound to perform *in vivo* studies and the pharmacodynamic properties of this compound are such that *in vivo* administration is not likely to be technically feasible; rather, the intent of our experiment is show efficacy *in vitro* as a proof-of-concept. However, we have now analyzed an additional number of canonical *MBNL1* targets and a second cell line (MV4;11) using semiquantitative PCR and demonstrate anticipated splicing changes following drug treatment (**Figure 5; line 283**).

Related to the above, it is not entirely clear if MBNL1 is required in only MLL-rearranged AML as claimed in the manuscript. Evaluation of genetic requirement for MBNL1 in non-MLL rearranged AML should be performed across a greater number of AML cell lines (only a single non-MLL rearranged cell line is tested in Figure 2). There are also public CRISPR data across AML cell lines which could be useful in this regard.

R1 response 6. The reviewer brings up a concern that too few cell lines were tested in our growth assays for the requirement of *MBNL1* expression. *MBNL1* is expressed nearly in all AML patient bone marrow samples but is upregulated selectively in MLL (**Figure 6A**). As noted, existing CRISPR liability data exists (David Sabatini screen - *Cell* **168**, 890-903 e15, 2017) which we now note indicates that 4 out of the 6 screen cell lines with MLL-translocations have the highest requirement of *MBNL1* out of the 15 cell lines tested (highest liability scores in OCI-AML2, MonoMac1, THP-1 and MOLM-13, respectively) (**Figure R1B**). Additionally, we introduced a new comparison of *MBNL1* knockdown splicing profiles compared to those in 481 ENCODE knockdown lines, over a dozen AML

genetically/cytogenetically defined subtypes and related hematopoietic cell populations, revealing specificity for *MBNL1* splicing in patients with MLL rearrangements (**Figure 6F**). To further bolster these results, we expanded the number of cell lines in which we tested the effect of *MBNL1*-knockdown. These results validate our prior findings that MLL rearranged cell lines are specifically susceptible to *MBNL* knockdown (**Figure 2B-G; line 135**). Further, in **Supplemental Figure 1A-C**, we have abstracted new public datasets (St. Jude/TARGET – PECAN Explorer; Beat AML – Tyner et al. 2018) with gene expression data by RNA-Seq and demonstrate high *MBNL1* expression in both pediatric and adult MLL-rearranged leukemias.

Is Mbn1 expression increased in the mouse MLL-AF9 retroviral model? It is very surprising that HOXA genes are not found as universally upregulated in MLL-rearranged leukemias in Figure 1. Can the authors comment on this?

R1 response 7. This is a valid question. Even within MLL-rearranged leukemias, *HOXA* gene expression exhibits variability (Starkova J, et al. *Pediatr Blood Cancer* 2010; Stam et al. *Blood* 2010, Trentin et al *Eur J Hematol* 2009) particularly within B-ALL. We have added this point to the manuscript (**Discussion, line 439**). Because Figure 1 attempts to capture distinguishing gene expression signatures across both AML and ALL, this likely explains the absence of *HOXA* genes in the intersection of all datasets. We also added a panel to **Supplemental Figure 1** showing that *Mbn1* is indeed highly expressed in the mouse retroviral MA9 leukemia model.

The “control” for MBNL1 expression in Figure 1B is not necessarily appropriate. How does the expression of MBNL1 in AML primary patient samples compare to normal adult CD34+ cells?

R1 response 8. We have updated **Figure 1B**; *MBNL1* expression is normalized to expression in CD34+ cord blood cells. Additionally, we further compare adult AML RNA-Seq and cord blood cell RNA-Seq data produced in the same laboratory (Guy Sauvageau) with the same protocol, hence, differences should not be due to laboratory or protocol effects (**Figure 6A**).

It appears from Figure 4G that Mbn1 is required for self renewal of normal mouse hematopoietic stem cells. What was the chimerism of HSPCs in primary transplant? Was a secondary competitive transplant attempted?

R1 response 9. While we did not have a sufficient number of appropriately aged *Mbn1*^{-/-} mice to perform primary and secondary transplants in the timeframe for our revisions, we were able to enumerate the size of the LT-HSC population (Lin⁻Sca1⁺Kit⁺ SLAM (CD150+CD41-CD48-)) between WT and KO mice (see **Figure 4I,J; line 249**). We believe that the decreased proportion of LT-HSCs likely accounts for the difference in engraftment and chimerism we observed in competitive transplants, as we transplanted the same number of Lin⁻ cells per mouse.

Targeting of RNA binding proteins is not a “new paradigm in targeted therapy.” This phrasing in the Abstract should be revised for accuracy. There have been numerous prior and ongoing efforts to therapeutically target RNA binding proteins in a variety of disorders (including very recent papers on targeting RNA binding proteins in AML).

R1 response 10. We have revised our language in the abstract (line 48)

Reviewer #2; computational splicing

1. I am surprised that the overlap of differential alternative splicing events found in patient samples and in *MBNL1* knockdown cells is low (only 85 events at the intersection of the two sets – Fig. 5D). Is this degree of overlap no different from or significantly higher than what would be expected by random chance? Do the authors have any explanation for why the overlap is low?

R2 response 1. We hypothesize that this limited number of common events is an artifact of comparing an experimental system (*MBNL1* knockdown in a cell line) against a large diverse patient dataset. In an attempt to control for this, both comparisons were coanalyzed, which allows us some control over methodological variation that would arise from asynchronous analyses. Furthermore, we elected to make the cytogenetically-normal/MLL-fusion comparison in patient samples in an attempt to control for the presence or absence of the fusion protein. Even this distinction will not perfectly isolate the effect of alternative splicing from the MLL-fusion; there is a high degree of heterogeneity within the CN cohort (e.g. spliceosome mutations) which could conceivably influence the results of our splicing analysis but is beyond the scope of this manuscript.

Here, we argue that alternative splicing concordance between MLL patients and cell lines is a more reliable measure of specificity for validation of common splicing programs. Here, concordance indicates how often a specific alternative splicing event in two comparisons occur in the same or opposite direction. Concordance values closer to 50% would indicate a random overlap, while concordance values of closer to 90% or 10% should indicate highly similar splicing signatures. An extremely high concordance was observed between both MLL-AF9 cell lines and short-hairpin designs (96-100%), at the level of overlapping splicing events. While concordance between shared splicing events was high within and between cell line *MBNL1* knockdowns, the specific overlap of alternative splicing events varied with the short hairpin used from 13% to 38%, depending on the comparison (see **Figure 6C**). In addition to high-concordance between the short hairpin or cell line, we also observe relatively high concordance between the patient MLL sample comparisons and the knockdown comparisons, ranging from 77-99% negative concordance. Negative rather than positive concordance is expected by the knockdown, as we predict that *MBNL1* mediates MLL-induced splicing, particularly loss of intron retention, which is partially rescued by *MBNL1* knockdown. Indeed, global assessment of *MBNL1* knockdown mediated splicing relative to over a dozen genetically/cytogenetically defined subtypes reveals specificity for *MBNL1* splicing in patients with MLL rearrangements (**Figure 6F**).

2. As related to the point above, the authors should perform motif scans of RBP/splicing factor motifs, to see if the motifs for certain splicing factors are enriched around the differential alternative splicing events found in patient samples, or in *MBNL1* knockdown cells, or in both. This may help answer why a large number of alternative splicing events are not shared between these two systems.

R2 response 2. We have now included motif enrichment (CisBP-RNA RNA recognition element enrichment analysis with HOMER) for known and *de novo* motifs, globally for both the MLL-rearranged samples (**Figure R1**) and for the *MBNL1* knockdowns (**Figure 6G, line 337**). While these analyses show a strong enrichment for *MBNL1* motifs, we have not relied on the presence or absence of such motifs as a reliable surrogate for *MBNL1* binding, as such motif searches are highly imperfect. The same issue exists for CLIP-Seq data, in which there is a high false negative prediction rate associated with those assays, depending on the cell system. Rather, we believe concordance of splicing events is a better surrogate for core splicing programs retained in the two systems that are dependent on *MBNL1* expression.

3. It is also somewhat surprising that intron retention is the most frequent type of alternative splicing events among the overlapped events, given that as a well-defined splicing factor MBNL1 should regulate many exon skipping events. I notice that the authors used AltAnalyze for the RNA-seq splicing analysis. Could this result be due to some inherent features of AltAnalyze? I think the authors should re-analyze their data using another popular software (for example, rMATS) to see if the breakdown between different types of alternative splicing events still holds.

R2 response 3. The reviewer brings up an important quality control question, which is how accurate our alternative splicing predictions relative to alternative approaches? This question is particularly important in the current study, in which variation in intron retention is specifically predicted to be decreased in the presence of MLL-oncofusions and rescued with MBNL1 knockdown. To address, below we provide specific rationale for this approach and benchmarking of the specific method applied in AltAnalyze to others, such as rMATS for intron retention and other modes of alternative splicing. Further, as noted in Reviewer #1 response 2, alternative intron retention is not observed with common splicing factor mutations using this algorithm, with the exception of *ZRSR2* mutants that are well-known to selectively mediate intron retention but are observed highly in MLL (Figure R1A). Finally, we note that performing the same alternative splicing analyses in the software MAJIQ, finds a similar rate of intron retention but few events (228 unique events out of 513 detected by MAJIQ and 406 out of 1084 events by AltAnalyze).

Here we applied a recently developed computational workflow developed in the Salomonis laboratory, MultiPath-PSI, implemented in the software AltAnalyze. MultiPath-PSI was previously described in a series of splicing focused malignancy and non-malignancy focused analyses, including its application to discover new subtypes of MDS in both deep bulk and single-cell RNA-Seq data (Muench et al. Blood, 2018 - PMC6251005, Fang et al. Cell Rep., 2018 - PMC5971064, Rindler et al. Sci Rep. 2017 - PMC5241776, Kamath-Rayne et al. BMC Med Genomics - PMC4619218). This algorithm was introduced in AltAnalyze version 2.1.1 and requires aligned BAM files as input (inclusion of novel junctions and strand prediction are recommended). Similar to other recent reported local splicing variation (LSV)-based methods, such as LeafCutter and MAJIQ, MultiPath-PSI considers the detected junctions within a restricted genomic interval for assessing local junction expression differences. Details on the benchmarking against rMATS, LeafCutter and MAJIQ and algorithm are provided at:

<https://altanalyze.readthedocs.io/en/latest/Algorithms/#multipath-psi-splicing-algorithm>.

These algorithms are constantly in flux, but our tool provides improved detection for such events over alternatives with simulation and qPCR validated data (**Figure R1C-E**).

In brief, unlike these alternative approaches, MultiPath-PSI considers all known and novel exon-exon junctions in a sample or cell and computes its relative detection compared to the local background of all genomic overlapping junctions that can be directly associated with the given gene of interest (sharing at least one known gene-associated splice-site). This calculation provides a more inclusive and conservative estimate of LSV (**Fig. R2A**). These same junctions are used to identify high confidence intron retention splicing events evidenced by pairs of exon-intron and intron-only mapping paired-end reads, sufficiently detected at both ends of a given intron (5' and 3'). This more stringent algorithm requires that only counts for exon-intron spanning reads are reported, in which multiple exon-intron spanning reads are detected at both ends of the intron and a matching paired-end read contained entirely within the intron is present (BAMtoExonBed module of AltAnalyze). PSI values are calculated for any junction with a minimum PSI difference of 0.1 (10% PSI) between any samples or cells. PSI values are only reported for samples or cells for a given

splicing event in which sufficient read-depth is present (minimum of 20 reads per examined junction interval). As denoted from a precision-recall analysis for simulation splicing (known, novel and intron retention) and qPCR defined splicing events, MultiPath-PSI has improved performance than alternative tested approaches (**Figure R2B,C**). This combined with the specificity of intron-retention predictions and alignment with prior described intron retention phenotypes in AML (**Figure R1A**), give us a high degree of confidence for the performance of these methods.

Reviewer #3; MLL models

1) Overall the bioinformatics is not well described. A) In the analysis presented in Fig 1A, what does the authors mean by “up regulated in MLLr AML/ALL”. Is this relative to other leukemic subtypes or relative to normal cells (if so which)? I failed to find any description of how this analysis was performed.

R3 response 1. We have clarified how this data was derived (**Methods, line 651**). In each study, gene expression signatures were derived which separated MLL-fusion leukemia from MLL-wt leukemias of different lineages. In our own analyses presented in Figure 6, induction of gene expression is to either cytogenetically normal AML patient samples or to an appropriate matched control where possible.

B) For the RNA-seq the authors identify commonly altered splicing events between patient (MLLr vs. other AMLs) and cell line (+/- KD of MBNL1). However, they don't describe what the frequency of these common events are (i.e. how many are common out of a total number of de-regulated events). This is especially important as they argue for the use of RNA splicing as a better predictor than gene expression. I suggest the authors make a Venn diagram for the splicing data similar to that in Figure 5D.

R3 response 2. We thank the reviewer for these recommendations and now provide details of the specific overlapping splicing events in **Figure 6D, E**. Details on the results of this analysis are provided above in response to Reviewer 2 (response 2).

2) The authors argue for a selective effect of MBNL1 inhibition in normal/non-MLLr versus MLLr cells. This is supported by cell growth and survival experiments in a human setting. Overall, I think the authors are over-interpreting the MLLr specificity of MBNL1. For cell line experiments, an important determinant is the growth rates and it is clear that the Kasumi-1 cell in Fig 2 are growing very slowly. To what extent is this true for the cells used in the inhibitor experiments, i.e. what are the relative growth rates of the cells used. Also, why are K562 and HL60 used for the inhibitor and Kasumi1 for the shRNA KD experiments?

R3 response 3: In the interests of consistency, we have performed additional experiments such that the same panel of cell lines are used in both the shRNA KD and inhibitor experiments (**Figure 2B-G, line 135**). In doing so, we show that even cells which are comparatively more proliferative (e.g. K562) MBNL1 knockdown does not impair growth, while also showing that MBNL1 is relatively dispensable for other oncofusions, such as *RUNX1-RUNX1T1* (Kasumi1) and *BCR-ABL* (K562).

3) The authors use an MBNL1 KO line to assess the in normal hematopoiesis and find relative mild effects. However, they do note a marked reduction in engraftment in a competitive transplantation setting suggestive of a stem cell defect. This could be tested by limited dilution

experiments and/or by FACS-based quantification of stem cell numbers.

The authors further argue that the relative mild delay in leukemic development in *MBNL1* KO mice could be due to the presence of *MBNL2*. This could be tested by performing shRNA KD experiments in this setting. The authors may also test the specific requirement for *MBNL1* in *MLLr* leukemias by conducting similar experiments using other leukemic fusion proteins. This would strengthen the argument for the specificity of *MBNL1* for *MLLr* leukemia.

R3 response 4. Regarding the reviewer's first point, we performed the first experiment as recommended (**Figure 4I, line 248**) – we found a decreased proportion of LSK-SLAM HSCs in *Mbnl1*^{-/-} mice, which likely accounts for the differences we observed in chimerism following competitive transplant.

We also attempted to formally test our hypothesis that *Mbnl2* may partially rescue the *Mbnl1*-knockout phenotype. Specifically, we performed a four-arm experiment, where we compared survival of mice transplanted with *Mbnl1*^{-/-} and *Mbnl1*^{+/+} retroviral MA9 leukemias, each with shNT and sh*Mbnl2* knockdowns. We also conducted a similar *in vitro* experiment with CFU assays. However, we could not obtain consistent differences in CFU number or separation of survival curves within the timeframe for our revisions. While there is still precedent for this observation of *Mbnl2* compensation in *Mbnl1* loss in the context of myotonic dystrophy models (Kuang-Yung, et al. *EMBO Mol Med* 2013), we were unable to demonstrate a similar phenomenon in our model, and as a result have tempered our language in the discussion.

As an alternative means to globally assess the role of *MBNL1* more broadly in AML and its specificity for *MLLr* leukemia, we obtained 481 RNA-binding protein (RBP) knockdown RNA-Seq datasets from ENCODE, defined over a dozen distinct genetically/cytogenetically defined subtypes of adult AML and analyzed 26 cell type-specific bulk RNA-Seq datasets to derive splicing profiles for the purpose of comparative evaluation. Comparison of *MBNL1* knockdown splicing differences among this large database identified the greatest negative concordance with *MLL* rearrangements (**Figure 6F**) as opposed to any other rearrangements (11% concordance). The second most negatively concordant AML subtype was *PML-RARA* which was only 31% concordant, indicating a weak level of splicing similarity with *MBNL1* knockdown. Several RBP knockdowns in K562 or HEPG2 cell lines were positively concordant with *MBNL1* knockdown in MOLM13, notably *MBNL1* knockdown in K562, *SRFB1*, *PCBP2*, *SF1* and *SRPK2*, suggesting potential collaborative functions or overlap in targets with the other unrelated factors. While AML patients with *IDH1* mutations (R132X) had an 80% concordance with *MBNL1* knockdown induced splicing, only a few splicing events were overlapping (n=21) to begin with. Hence, these analyses show that *MBNL1* splicing in AML is most correlated with *MLL*-splicing and is reproducible with other *MBNL1* knockdowns in different cell lines (MV411 and K562).

Minor point

1) The authors use *MLL-AF9 Tet-off human CD34+* cells but do not provide a description of their generation or origin. This should be included.

R3 response 5. We have clarified this in the Methods section (**line 528**)

Reviewer Figures

Figure R1. Additional Evidence for Repression of Intron-Retention in MLL Mediated by *MBNL1*. A) Relative ratio of alternative exon inclusion (left) and exon inclusion (right) for distinct categories of alternative splicing or alternative promoter regulation in different adult AML subtypes (Leucegene cohort). Alternative splicing events were quantified by MultiPath-PSI as described (Methods). B) Statistical enrichment of RNA recognition elements from the CisBP-RNA database with the software HOMER for alternatively spliced exons/introns (Z-score) and *MBNL1* gene expression in MLL rearranged patient samples versus cytogenetically normal matched AMLs. C) CRISPR screen scores (CSS) from a published database of leukemic growth liabilities in 13 screened AML cell lines (*Cell* **168**, 890-903 e15, 2017). Cell lines in red are those containing MLL rearrangements.

Figure R2. Reproducibility and Performance of MultiPath-PSI for Alternative Splicing Quantification. A) MultiPath-PSI example splicing event calculation for intron retention (H) or more complex use cases (F), from exon-exon and exon-intron junction read count ratios. B) Previously reported delta PSI values for significantly differential RT-PCR splicing events were compared to MultiPath-PSI splicing events for 8 mouse liver and 8 mouse cerebellum samples, indicating high concordance. C) Precision and recall evaluation of MultiPath-PSI relative to three high-confidence local splicing variation approaches using simulated data. Simulated RNA-Seq data was produced in the software polyester from full-length known and novel isoforms derived from Cufflinks. As an unbiased evaluation of possible known and novel alternative splicing events, we developed a simulated RNA-Seq dataset derived from known and novel isoform predictions for pluripotent stem cells (PSC) and in vitro derived day 30 cardiomyocytes (CM) (<https://www.synapse.org/#!/Synapse:syn12104338>). To create this simulation dataset, we first predicted known and novel isoforms and expression estimates using Cufflinks on biological triplicate samples (PSC and CM). Additional details are available online at: <https://altanalyze.readthedocs.io/en/latest/Algorithms/#multipath-psi-splicing-algorithm>. D) AUCPR was calculated using junctions mutually detected by each specific application and the synthetic ground state truth (see Filtered AUCPR) or all synthetic junctions (All AUCPR). TP=true positive synthetic events. Note the term “events” in the table is used to represent unique splice-junction clusters instead of individual splicing events.

Reviewers' comments:

Reviewer #1 (Remarks to the Author):

The authors have fully addressed my initial questions and concerns.

Reviewer #2 (Remarks to the Author):

I am not convinced that the authors have successfully addressed my major concern for this work. As raised in my previous review, the overlap of differential alternative splicing events found in patient samples and in MBNL1 knockdown cells is low. This observation is reinforced by Fig. 6D and Fig. 6E of the revised manuscript, which now show data from MBNL1 knockdown using two different shRNAs in two different cell lines. The authors proposed a new "concordance of alternative splicing events" metric and argued that this metric indicates that the overlapping differential alternative splicing events display concordant directions of change in patient samples and in MBNL1 knockdown cells. However, this "concordance" metrics seems dubious and lacks proper justification/validation, and the issue remains that the vast majority (over 95%) of the differential splicing events in patient samples didn't change upon MBNL1 knockdown. With the data presented, MBNL1 could play a role in regulating a set of AS events in MLL-rearranged leukemia, but the role appears minor.

Reviewer #3 (Remarks to the Author):

I believe the authors have addressed my concerns adequately in the revised version of the manuscript.

We wish to thank our reviewers again for your constructive criticism and thoughtful feedback regarding our manuscript “MBNL1 regulates essential alternative RNA splicing patterns in MLL-rearranged leukemia” (NCOMMS-19-02864B.) Please find attached a further revised version of our manuscript with additional experimental data addressing the critiques raised by our reviewers. Our responses to their specific critiques are appended below.

Reviewer #2

I am not convinced that the authors have successfully addressed my major concern for this work. As raised in my previous review, the overlap of differential alternative splicing events found in patient samples and in MBNL1 knockdown cells is low. This observation is reinforced by Fig. 6D and Fig. 6E of the revised manuscript, which now show data from MBNL1 knockdown using two different shRNAs in two different cell lines. The authors proposed a new “concordance of alternative splicing events” metric and argued that this metric indicates that the overlapping differential alternative splicing events display concordant directions of change in patient samples and in MBNL1 knockdown cells. However, this “concordance” metrics seems dubious and lacks proper justification/validation, and the issue remains that the vast majority (over 95%) of the differential splicing events in patient samples didn't change upon MBNL1 knockdown. With the data presented, MBNL1 could play a role in regulating a set of AS events in MLL-rearranged leukemia, but the role appears minor.

R2 Response 1:

The reviewer highlights a challenge within the field of leukemia, which is how to properly assess the *in vivo* impact of genomic alternations observed *in vitro*, and more specifically, how to assess the relative global impact of splicing alternations in patients compared to those observed in leukemia cell lines. Alternative splicing changes *in vivo* are likely to be associated with both leukemic blasts but also niche-associated bone marrow influences, which cannot be effectively and accurately modeled *in vitro*. To more clearly demonstrate this phenomenon, we performed a meta-analysis of existing well-described *in vitro* models of human leukemias. Examination of alternative splicing events in the Leucegene¹ cohort from patients with SRSF2-P95 mutations to those an engineered cell line (K562) with the same mutation (Zhang J, et al. PNAS, 2015²), finds that less than 10% of patient identified specific events overlap with their *in vitro* analogues (Figure R1A). Nonetheless, *in vitro* modeling of SRSF2 mutations have been well demonstrated using such a model to recapitulate observed patient splicing defects as cited above. We note that this analysis as visualized in Figure R1A uses the same alternative splicing analysis workflow as that used in our own analysis of *MBNL1* knockdown and MLL-rearranged primary patient data presented in our manuscript. Similarly, less than 5% of *MLL*-fusion patient splicing events overlap with those identified in cell lines with MLL-fusions versus those without such fusions from the Cancer Cell Line Encyclopedia (CCLE) project (deep paired-end RNA-Seq data in 7 distinct MLL leukemia cell lines versus 19 control non-*MLL* rearranged AML cell lines) (Figure R1B). Nonetheless, such *MLL* cell lines are considered reliable models for *MLL*

gene regulatory and essentiality modeling in the leukemia field at large.³⁻⁶ Additionally, our *in vivo* analyses of *MLL* patients (Leucegene) demonstrate wide-spread splicing changes relative to other genomic lesions, raising the possibility that the reason for such a proportionally small overlap should not be attributed purely to a poor splicing phenotype. Both of these results are in line with our observation of a relatively low overlap between any *in vitro* and *in vivo* models from our own knockdown analysis (Figure R1B). In addition to *in vitro* and *in vivo* differences, there will always be unavoidable technical differences in RNA-Seq protocols, instruments, sample genetics and sample preparation that are likely to further confound these comparisons.

However, when we examine the concordance of splicing signatures identified between these *in vitro* and *in vivo* studies, we do find that *in vitro* models produce highly concordant splicing signatures when correlated to corresponding *in vivo* comparisons, with a similar extent to that observed in our MBNL1 knockdown studies (Figure R1C). This analysis was performed using *MLL*-fusion CCLE, SRSF2-P95 and two U2AF1-S34 engineered cell lines, with similar outcomes. Hence, the specificity of our splicing results are comparable to prior studies. These data showing high concordance over absolute number of common events argue for an evaluation of splicing specificity over sensitivity, which our concordance analysis addresses. We previously used this same type of concordance analysis to uncover HNRNPK as a splicing regulator in TGF-beta active *MLL* patients, which was subsequently evidenced in a murine model.⁷

We attempted to summarize this discussion in the manuscript by elaborating on the criteria used to determine concordance between alternatively spliced features (lines 324-335), and further examining individual feature-level differential splicing to visually confirm the shared direction of changes predicted by our analysis (Supplemental Figure 6.)

Figure R1. Concordance of *in vitro* derived splicing signatures accurately reflect patient derived splicing. Alternative splicing differences are compared from prior published *in vitro* models of AML splicing for both relative overlap and concordance of splicing events (percentage of overlapping events regulated in the same direction). A) Venn Diagram demonstrating the small percentage of overlap of detected alternative splicing events determined from Leucegene patients with *SRSF2*-P95X mutations versus patients without splicing factor mutations and those from a CRISPR/Cas9 edited leukemia cell line (K562) with the *SRSF2*-P95H mutation versus WT K562 cells ($n=4$)². Differential splicing events were calculated in the software MultiPath-PSI ($p < 0.01$), following alignment of the FASTQ files to the human genome (hg19) with the software STAR. B) Similar low percentage overlap of alternative splicing events in Leucegene *MLL*-fusion patients versus cytogenetically normal AML and Cancer Cell Line

Encyclopedia (CCLE) MLL-fusion containing leukemia cell lines versus non-MLL cell lines. C) Relatively high concordance for cell line alternative splicing, when comparing distinct in vitro introduced myeloid malignancy mutations (*SRSF2*-P95², *U2AF1*-S34F⁸) or patient derived rearrangements (*MLL*) to a database of alternative splicing signatures from known AML genomic variants (Leucegene¹). In vitro derived alternative splicing events for *SRSF2*-P95H and *MLL*, correspond to those from panels A and B respectively, while *U2AF1*-S34F alternative splicing events were derived from CRISPR/Cas9 edited HBEC3kt cells⁸ or over-expressed *U2AF1*-S34F in erythroid progenitors⁹. Note, these analyses demonstrate that both concordance and the total number of overlapping splicing events together are the most reliable indicator of in vivo/in vitro splicing similarity.

References

- 1 Lavalley, V. P. *et al.* The transcriptomic landscape and directed chemical interrogation of MLL-rearranged acute myeloid leukemias. *Nat Genet* **47**, 1030-1037, doi:10.1038/ng.3371 (2015).
- 2 Zhang, J. *et al.* Disease-associated mutation in SRSF2 misregulates splicing by altering RNA-binding affinities. *Proc Natl Acad Sci U S A* **112**, E4726-4734, doi:10.1073/pnas.1514105112 (2015).
- 3 Borkin, D. *et al.* Pharmacologic inhibition of the Menin-MLL interaction blocks progression of MLL leukemia in vivo. *Cancer Cell* **27**, 589-602, doi:10.1016/j.ccell.2015.02.016 (2015).
- 4 Sun, Y. *et al.* HOXA9 Reprograms the Enhancer Landscape to Promote Leukemogenesis. *Cancer Cell* **34**, 643-658 e645, doi:10.1016/j.ccell.2018.08.018 (2018).
- 5 Volk, A. *et al.* A CHAF1B-Dependent Molecular Switch in Hematopoiesis and Leukemia Pathogenesis. *Cancer Cell* **34**, 707-723 e707, doi:10.1016/j.ccell.2018.10.004 (2018).
- 6 Erb, M. A. *et al.* Transcription control by the ENL YEATS domain in acute leukaemia. *Nature* **543**, 270-274, doi:10.1038/nature21688 (2017).
- 7 Muench, D. E. *et al.* SKI controls MDS-associated chronic TGF-beta signaling, aberrant splicing, and stem cell fitness. *Blood* **132**, e24-e34, doi:10.1182/blood-2018-06-860890 (2018).
- 8 Fei, D. L. *et al.* Wild-Type U2AF1 Antagonizes the Splicing Program Characteristic of U2AF1-Mutant Tumors and Is Required for Cell Survival. *PLoS Genet* **12**, e1006384, doi:10.1371/journal.pgen.1006384 (2016).
- 9 Yip, B. H. *et al.* The U2AF1S34F mutation induces lineage-specific splicing alterations in myelodysplastic syndromes. *J Clin Invest* **127**, 2206-2221, doi:10.1172/JCI91363 (2017).

REVIEWERS' COMMENTS:

Reviewer #2 (Remarks to the Author):

I appreciate the authors' thoughtful responses to my remaining concern. The manuscript is now ready for publication.